# Effect of chemical composition on the electrical conductivity of
# gneiss at high temperatures and pressures
Lidong Dai[1,*], Wenqing Sun[1,2], Heping Li[1], Haiying Hu[1], Lei Wu[1], and Jianjun Jiang[1]
[1]Key Laboratory of High-Temperature and High-Pressure Study of the Earth's Interior,
Institute of Geochemistry, Chinese Academy of Sciences, Guiyang, 550081, China
[2]University of Chinese Academy of Sciences, Beijing, 100049, China
*Correspondence to*: Lidong Dai (dailidong@vip.gyig.ac.cn)
**Abstract.** Electrical conductivity of gneiss samples with different chemical
compositions ($W_A$=Na$_2$O+K$_2$O+CaO=7.12%, 7.27% and 7.64% weight percent) were
measured using a complex impedance spectroscopic technique at 623–1073 K and 1.5
GPa and a frequency range of $10^{-1}$ to $10^6$ Hz. Simultaneously, a pressure effect on the
electrical conductivity was also determined for the $W_A$=7.12% gneiss. The results
indicated that the gneiss conductivities markedly increase with total alkali and calcium
ion content. The sample conductivity and temperature conform to an Arrhenius
relationship within a certain temperature range. The influence of pressure on gneiss
conductivity is weaker than temperature, although conductivity still increases with
pressure. According to various ranges of activation enthalpy (0.35–0.52 eV and 0.76–
0.87 eV) at 1.5 GPa, two main conduction mechanisms are suggested that dominate the
electrical conductivity of gneiss: impurity conduction in the lower temperature region
and ionic conduction (charge carriers are K$^+$, Na$^+$ and Ca$^{2+}$) in the higher temperature
region. The electrical conductivity of gneiss with various chemical compositions cannot
be used to interpret the high conductivity anomalies in the Dabie–Sulu ultrahigh
pressure metamorphic belt. However, the conductivity–depth profiles for gneiss may
provide an important constraint on the interpretation of field magnetotelluric
conductivity results in the regional metamorphic belt.

**1 Introduction**

According to magnetotelluric (MT) and geomagnetic depth sounding results, electrical conductivity of geological samples at high temperature and pressure can be used to extrapolate the mineralogical composition and thermodynamic state in the Earth's interior (Maumus et al., 2005; Dai et al., 2008; Hui et al., 2015; Manthilake et al., 2015; Li et al., 2016; Hu et al., 2017). High conductivity anomalies are widely distributed in the middle to lower crust and upper mantle, and there are various causes of these anomalies in different regions (Xiao et al., 2007, 2011; Pape et al., 2015; Novella et al., 2017). Hence, it is crucial to comprehensively measure the electrical conductivities of minerals and rocks that are distributed in the deep Earth. A series of electrical conductivity results of the main minerals and rocks have been reported in previous studies under high temperature and pressure conditions (Fuji-ta et al., 2007; Hu et al., 2011, 2013; Dai et al., 2012; Yang et al., 2012; Sun et al., 2017a). However, electrical conductivity of most metamorphic rocks have not been explored at high temperature and pressure, and thus the interpretation of high conductivity anomalies distributed in representative regional metamorphic belts is still not comprehensive.

A regional metamorphic ultrahigh pressure belt for the Dabie–Sulu orogen is a complexly giant geotectonic unit in central-eastern China. Geophysical exploration results confirmed that a large number of high conductivity anomalies have been observed in metamorphic belts (Xiao et al., 2007; Wannamaker et al., 2009; Zeng et al., 2015). Metamorphic rocks (e.g., slate, schist, gneiss, granulite and eclogite) with different degrees of metamorphism play an important role because of their widespread distribution in regional metamorphic belts. Dai et al. (2016) measured the electrical conductivity of dry eclogite at 873–1173 K, 1.0–3.0 GPa and different oxygen partial pressures (using Cu+CuO, Ni+NiO and Mo+MoO$_2$ solid oxygen buffers), and found that the hopping of small polaron is the dominant conduction mechanism for dry eclogite at high temperature and pressure. The electrical conductivity of natural eclogite is much lower than the high conductivity anomaly in the Dabie–Sulu ultrahigh pressure

metamorphic (UHPM) belt of eastern China. Granulite is another important
metamorphic rock distributed in a majority of regional metamorphic belts. The
electrical conductivity of granulite is lowered by repetitive heating cycles with a
conductivity range about $10^{-7}$–$10^{-2}$ S/m at 1.0 GPa and up to about 900 K (Fuji-ta et al.,
2004). Due to the complicated mineralogical assemblage of granulite and rock structure,
the features of the electrical conductivity values over heating cycles have not been
explained, and the conduction mechanism for granulite not definitively stated. Gneiss
is formed at middle to lower crustal pressure and temperature conditions, and widely
distributed in regional metamorphic belts. The main rock-forming minerals of gneiss
are feldspar, quartz and biotite. The electrical conductivity of gneiss increases with
temperature, and the conductivity values range from about $10^{-4}$–$10^{-2}$ S/m at up to 1000
K and 1.0 GPa (Fuji-ta et al., 2007). On the basis of the dominant rock-bearing
mineralogical assembly of the metamorphic rock, gneiss can generally be divided into
types, such as plagioclase gneiss, quartz gneiss and biotite gneiss. Therefore, it is crucial
to investigate the electrical conductivity of gneisses with various chemical
compositions and mineralogical constituents. The electrical conductivity of granite
dramatically increases with alkaline and calcium ion content at 623–1173 K and 0.5–
1.5 GPa (Dai et al., 2014). Impurity conduction has been proposed to be the dominant
conduction mechanism for granite in the lower temperature region, and alkaline ions,
including $K^+$, $Na^+$ and $Ca^{2+}$, are probable charge carriers at higher temperatures.
In the present study, we measured the electrical conductivity of gneiss samples in
situ under 0.5–2.0 GPa, 623–1073 K and three different chemical compositions. The
influences of temperature, pressure and chemical composition on the gneiss electrical
conductivity were determined, and the dominant conduction mechanism for gneiss is
discussed in detail. On the basis of the conductivity results, the geophysical implications
for the high conductivity anomalies of the Dabie–Sulu UHPM belt were explored in
depth.




## 2 Experimental procedures



### 2.1 Sample preparation


Three relatively homogeneous natural gneiss samples with parallel to foliation direction
were collected from Xinjiang, China. The sample surfaces were fresh, non-fractured
and non-oxidized, without evidence of alteration before and after the experiments. To
determine the gneiss mineralogical assemblage, we used optical microscopy and
scanning electron microscopy (SEM) at the State Key Laboratory of Ore Deposit
Geochemistry, Institute of Geochemistry, Chinese Academy of Sciences, Guiyang,
China. The major elemental content of the gneiss samples was analyzed by X-ray
fluorescence spectrometry (XRF) at Australian Laboratory Services, Shanghai, China.
The main rock-forming minerals of three gneiss samples were feldspar, quartz and
biotite (Fig. 1). The volume percentage varied for each corresponding rock-forming
mineral in different gneiss samples (Table 1). Three gneiss samples had the same
mineralogical assemblage, and all of them belong to the biotite-bearing felsic gneiss.
Table 2 shows the results of whole rock analysis by XRF for the three gneiss samples.
We found that the total alkali, such as $K^+$ and $Na^+$, and the divalent cationic calcium
metal ion content varied for each sample (Table 2).

### 2.2 Impedance measurements



High temperatures and pressures for the experiments were generated in a YJ-3000t
multi-anvil apparatus, and the impedance spectra were collected using a Solartron-1260
impedance/gain-phase analyzer at the Key Laboratory of High-Temperature and High-
Pressure Study of the Earth's Interior, Institute of Geochemistry, Chinese Academy of
Sciences, Guiyang, China. All components of the experimental assemblage (ceramic
tubes, pyrophyllite, $Al_2O_3$ and MgO sleeves) were previously baked at 1073 K for 12 h
in a muffle furnace to avoid the influence of absorbed water on the electrical
conductivity measurements. The sample was then loaded into a MgO insulation tube
(Fig. 2). Two nickel disks (6.0 mm in diameter and 0.5 mm in thickness) were applied
to the top and bottom of sample to act as electrodes. To shield against external
electromagnetic and spurious signal interference, a layer of nickel foil with a thickness
of 0.025 mm was installed between the alumina and magnesia sleeves. These sleeves
have good insulating properties for current and transmitting pressure. A pyrophyllite
cube (edge length: 32.5 mm) was used as the pressure medium, and the heater was
composed of three-layer stainless steel sheets with a total thickness of 0.5 mm. The
sample assembly was placed in an oven at 330 K to keep it dry before the experiment.
In the experiments, the pressure was slowly increased to the desired value at a rate
of 1.0 GPa/h, and then the temperature was increased at a rate of 300 K/h to the
designated values. The Solartron-1260 impedance/gain-phase analyzer with an applied
voltage of 3 V and frequency range of $10^{-1}$–$10^6$ Hz was used to collect impedance
spectra when the pressure and temperature were stable. At the desired pressure, the
spectra were measured at a certain temperature, which was changed in 50 K intervals.
The impedance spectra of gneiss samples with $W_A$ ($Na_2O+K_2O+CaO$)=7.12% were
collected under conditions of 0.5–2.0 GPa and 623–1073 K. The spectra of the other
two gneiss samples ($W_A$=7.27% and 7.64%) were measured at 623–1073 K and 1.5
GPa. To confirm the data reproducibility, we measured the electrical conductivity of
gneiss over two heating and cooling cycles at a constant pressure. The errors of
temperature and pressure were ±5 K and ±0.1 GPa, respectively.

**3 Results**

The typical complex impedance spectra for the run DS12 gneiss samples at 1.5 GPa
and 623–1073 K are shown in Figure 3. All of these obtained spectra are composed of
an almost ideal semicircle in the high frequency domain and an additional tail in the
lower frequency domain. Other complex impedance spectra of the gneiss samples at
other pressures displayed the same characteristics of those shown in Figure 3. Figure 4
displays the real and imaginary parts of complex impedance for the runs DS13 and
DS14 gneiss samples as a function of the measured frequency at 1.5 GPa and 623–1073
K. The real part values almost remain unchanged over a frequency range of $10^6$–$10^4$ Hz,
and sharply increased at $10^4$–$10^2$ Hz; these values then slowly increased within the $10^2$
to $10^{-1}$ Hz lower frequency region. The values of imaginary parts almost remain
unchanged within a frequency range of $10^6$–$10^5$ Hz, the values gradually increased at
$10^5$–$10^3$ Hz and decreased at $10^3$–$10^1$ Hz; and these values then slowly increased in the
$10^1$ to $10^{-1}$ Hz lower frequency region. Roberts and Tyburczy (1991) and Saltas et al.
(2013) have suggested that the ideal semicircle represents the bulk electrical properties
of a sample, and the additional tail is characteristic of diffusion processes at the sample–
electrode interface. Hence, the bulk sample resistance can be obtained by fitting the
ideal semicircle in the high frequency domain. A series connection of resistance and
constant phase elements ($R_S$–$CPE_S$) and the interaction of charge carriers with the
electrode ($R_E$–$CPE_E$) was applied to be the equivalent circuit. All fitting errors of the
electrical resistance were less than 5%. Based on the sample size and electrical
resistance, the electrical conductivity of the sample was calculated:
$$\sigma = L / SR \tag{1}$$

where $L$ is the height of the sample (m), $S$ is the cross-sectional area of the electrodes
($m^2$), $R$ is the fitting resistance ($\Omega$) and $\sigma$ is the electrical conductivity of the sample
(S/m).
The logarithmic electrical conductivities of the gneiss samples were plotted
against the reciprocal temperatures under conditions of 623–1073 K and 0.5–2.0 GPa.
The electrical conductivities of gneiss with $X_A$=7.12% were measured in two sequential
heating and cooling cycles at 1.5 GPa (Fig. 5). After the first heating cycle, electrical
conductivities of the gneiss at the same temperature were close to each other in other
cycles. We confirmed that our experimental data were reproducible, and the gneiss
sample was kept at a steady state after the first heating cycle. Two different linear
relationships of logarithmic electrical conductivity and reciprocal temperature were
separated by an inflection point. The electrical conductivity of gneiss with $W_A$=7.12%
significantly increased with temperatures above 723 K at 0.5–1.0 GPa, and this
phenomenon occurred after 773 K at 1.5–2.5 GPa (Fig. 6). The electrical conductivity
of the samples increased with pressure, but the effect of pressure on conductivity was
weaker than temperature. For other gneiss samples ($W_A$=7.27% and 7.64%), the
inflection points appeared at 773 K under all designated pressures (Fig. 7). In a specific
temperature range, the relationship between electrical conductivity and temperature fits
the Arrhenius formula:

$$\sigma = \sigma_0 \exp(-\Delta H / kT)$$ (2)
$$\Delta H = \Delta U + P\Delta V$$ (3)

where $\sigma_0$ is the pre-exponential factor (S/m), $\Delta H$ is the activation enthalpy (eV), $k$ is the
Boltzmann constant (eV/K), $T$ is the absolute temperature (K), $\Delta U$ is the activation
energy (eV), $P$ is the pressure (GPa) and $\Delta V$ is the activation volume (cm$^3$/mole). All
fitting parameters for the electrical conductivities of three gneiss samples are listed in
Table 3. The activation enthalpy values ($\Delta H$) for the gneiss samples are 0.35–0.58 eV
in the lower temperature region and 0.71–1.05 eV in the higher temperature region,
respectively. In addition, the logarithms of pre-exponential factor values (Log $\sigma_0$) were
transformed from negative to positive from the correspondent lower to the higher
temperature ranges.

The total alkali and calcium ion content of $K_2O$, $Na_2O$ and CaO is a remarkable

influence on the electrical conductivities of the gneiss samples. As shown in Figure 7,
the electrical conductivity of the gneiss samples increased with the total weight percent
of $K_2O$, $Na_2O$ and CaO. It reflects the fact the electrical conductivity of the gneiss
samples is controlled mainly by minerals that contain abundant $K_2O$, $Na_2O$ and CaO.
The cations of feldspar are $K^+$, $Na^+$ and $Ca^{2+}$, and $K^+$ is also the main cation of biotite.
Furthermore, impurity ions ($K^+$, $Na^+$ and $Al^{3+}$) have been suggested to be the charge
carriers for quartz samples (Wang et al., 2010). In addition, the electrical conductivity
of the gneiss samples do not regularly change with variations in biotite-bearing content
(Fig. 7 and Table 1). Based on all of the experimental results, the biotite content is not
the main influential factor on the electrical conductivity of gneiss. Therefore, we cannot
distinguish the specific mineral that controls the electrical conductivity of the gneiss
samples. However, it was reasonable to consider the gneiss sample as a complex whole,
and analyze the electrical conductivity of gneiss with various chemical compositions at
high temperature and pressure.

**4 Discussions**

**4.1 A comparison with previous studies**

As three constituent minerals of gneiss, feldspar, biotite and quartz dominated the
electrical conductivity of the whole rock at high temperature and pressure. Due to their
sophisticated mineralogical assemblage and rock structure, the gneiss samples were
unstable in the first heating cycle. In this process, the impurity ions may have been
distributed, the grain size slightly changed and the microfractures gradually closed.
After the first cycle, the electrical conductivity of the gneiss samples had good
repeatability. This suggested that the gneiss samples were in a stable state. The electrical
conductivity range of the gneiss samples with various chemical compositions was about
$10^{-5}$–$10^{-1}$ S/m at 623–973 K and 0.5–2.0 GPa. The electrical conductivity was slightly
related to pressure, and conforms to previous conclusions that the influence of pressure
on mineral and rock conductivity is much weaker than temperature (Xu et al., 2000; Hu
et al., 2011; Dai and Karato, 2014a, b). The possible reason is that the effect of pressure
on the activity of the charge carriers is weaker than temperature. The total alkaline ion
content of $K_2O$, $Na_2O$ and CaO has crucial influence on the electrical conductivity of
gneiss. Previous studies have investigated the electrical conductivity of minerals and
rocks with various chemical compositions, and the conclusions were similar to ours
(Dai et al., 2014). Fiji-ta et al. (2007) performed the electrical conductivity of gneiss
perpendicular and parallel to foliation at up to 1000 K and a constant pressure of 1.0
GPa. The conductivity of gneiss measured perpendicular to foliation was one order of
magnitude lower than the value measured parallel to foliation. However, the influence
of pressure and chemical composition on the electrical conductivity of gneiss has not
been studied. In the present work, we investigated the electrical conductivity of gneiss
parallel to foliation. As shown in Figure 8, the electrical conductivity of gneiss from
Fuji-ta et al. (2007) was higher than our results in the lower temperature range, and
values were lower than the conductivity of gneiss with $W_A$= 7.27% and 7.64% in this
study. This discrepancy is probably caused by varying chemical compositions of the
gneiss samples. Dai et al. (2014) measured the electrical conductivity of granite at 0.5–
1.5 GPa and 623–1173 K, and the main rock-forming minerals were also quartz,
feldspar, and biotite. They found that the content of calcium and alkali ions significantly
affected the electrical conductivity of granite under conditions of high temperature and
high pressure. Electrical conductivities of granite and gneiss increased with calcium
and alkali ion content. However, the electrical conductivity of granite was much lower
than gneiss (Fig. 8). This difference may be caused by the various chemical
compositions and rock structures between granite and gneiss. Feldspars are main
constituent rock-forming minerals in gneiss, and thus it is important to compare the
electrical conductivity of feldspar. The electrical conductivity of K-feldspar is one order
of magnitude lower than albite, and $K^+$ and $Na^+$ ions are the charge carriers of K-
feldspar and albite, respectively (Hu et al., 2013). As shown in Figure 8, the electrical
conductivity of alkali feldspar is much higher than the gneiss samples. This may be
because the concentration of alkali ions in feldspar is higher than gneiss. In addition,
granulite is another significant metamorphic rock, and usually coexists with gneiss. The
electrical conductivity of granulite is moderately higher than gneiss. The electrical
conductivity of quartz at 1.0 GPa is slightly lower than gneiss with $X_A$=7.27% at 1.5
GPa, and the slope of the linear relationship between the logarithm of electrical
conductivity and the reciprocal of temperature for quartz is close to gneiss in a lower
temperature range (Wang et al., 2010). The conductivity of phlogopite is higher than
gneiss with $X_A$=7.64% at higher temperatures (>773 K), and lower than gneiss samples
at lower temperatures (<773 K). Furthermore, the slope of the linear relationship
between the logarithm of electrical conductivity for the phlogopite and the reciprocal
temperature is much higher than the slope of the gneiss samples (Li et al., 2016).
Compared with Ferri et al. (2013), the electrical conductivity of the garnet–biotite–
sillimanite residual enclave (JOY2-X4) was very close to our conductivity results for
the run DS13 and DS14 gneiss samples in the lower temperature and higher temperature
regions, respectively. The electrical conductivity of sample JOY2-X4 was slightly
lower than the run DS12gneiss sample. In addition, the electrical conductivity of natural
metapelite (PP216) from Hashim et al. (2013) was close to the values of the run DS12
gneiss sample in the lower temperature region, and the slope between logarithmic
conductivities and reciprocal temperature for the PP216 metapelite was higher than the
gneiss samples in the higher temperature region.

**4.2 Conduction mechanism**

The logarithm of electrical conductivities and reciprocal temperatures showed linear
relationships at the lower and higher temperature ranges, respectively. This implies that
the dominant conduction mechanism for our gneiss samples in the lower temperature
range is different from the higher temperature range. The mineral assemblage and
chemical composition of gneiss samples are very complicated, and thus the conduction
mechanisms for gneiss samples are difficult to determine. Feldspar, quartz and biotite
are the dominant minerals in the gneiss samples. Previous studies have suggested that
the conduction mechanism for feldspar minerals is ionic conduction and the charge
carriers are $K^+$, $Na^+$ and $Ca^{2+}$ (Hu et al., 2013). The conduction mechanism for biotite
has not been studied, whereas the charge carriers of phlogopite were proposed to be $F^-$
and $K^+$ (Li et al., 2016). For quartz, the conduction mechanism was impurity ionic
conduction, and the dominant charge carriers migrate by moving the alkali ions in
channels (Wang et al., 2010). Therefore, we deduced that the conduction mechanism
for gneiss samples may be related to ions. The activation enthalpy is one crucial
evidence for the conduction mechanism of minerals and rocks (Dai et al., 2016). The
activation enthalpies for gneiss samples are 0.35–0.58 eV in the lower temperature
region, and 0.77–0.87 eV in the higher temperature region (Table 3). Dai et al. (2014)
studied the electrical conductivity of granite that had the same mineralogical
assemblage as the gneiss samples. They proposed that the conduction mechanism at the
lower temperature range was the impurity conduction owing to the low activation
enthalpy (0.5 eV), whereas the mechanism was ionic conduction with a high activation
enthalpy (1.0 eV) at the higher temperature range. The activation enthalpy for gneiss
was close to the values for granite at the lower and higher temperature ranges. The
activation enthalpies for albite and K-feldspar were 0.84 and 0.99 eV, respectively (Hu
et al., 2013). With increasing pressure, the electrical conductivity of gneiss increased
accordingly. The activation volumes for one gneiss sample (DS12) were -7.10 $cm^3$/mole
and -2.69 $cm^3$/mole in the low and high temperature regions, respectively. We can
compare gneiss with the electrical conductivity of eclogite, another representative
metamorphic rock. Recently, Dai et al. (2016) measured the electrical conductivity of
dry eclogite and the negative activation volume for eclogite was -2.51 $cm^3$/mole under
1.0–3.0 GPa and 873–1173 K. It was proposed that the main conduction mechanism for
dry eclogite is intrinsic conduction (Dai et al., 2016). In addition, Figure 7 shows that
the increasing content of alkali and calcium ions significantly enhances the electrical
conductivity of gneiss samples. Therefore, the impurity conduction (possible charge
carriers: $K^+$, $Na^+$, $Ca^{2+}$ and $H^+$) and ionic conduction (possible charge carriers: $K^+$, $Na^+$
and $Ca^{2+}$) are suggested to be the conduction mechanisms at lower and higher
temperature ranges, respectively.

**4.3 Effect of chemical composition on electrical conductivity**

The influence of chemical composition ($Na_2O+K_2O+CaO$) on the electrical
conductivity of the gneiss samples was very significant, as shown in a previous study
that the electrical conductivity of granite samples is closely related to the alkali and
calcium ion content (Dai et al., 2014). The electrical conductivity of granite samples at
high temperature and pressure can be fitted as a function of ($Na_2O+K_2O+CaO$)/$SiO_2$
(Dai et al., 2014). However, the electrical conductivity of gneiss samples does not
regularly change with variations in ($Na_2O+K_2O+CaO$)/$SiO_2$. This may be due to the
more complicated mineralogical assemblage and chemical composition of gneiss: for
mineralogical assemblage, the biotite content of the gneiss sample is higher than granite;
as for the chemical compositions, the contents of $SiO_2$ for gneiss are lower than those
of granite samples; the contents of the calc-alkali ions are approximate between gneiss
and granite samples. Hu et al. (2013) demonstrated that the electrical conductivity of
alkali feldspar significantly depends on the value of Na/(Na+K). This suggests that the
electrical conductivity of gneiss is affected by the total content of alkali and calcium
ions, as well as the ratios between various ions.

**5 Geophysical implications**

As a typical metamorphic rock in the present research region, gneiss is widespread in
the UHPM zone (Zheng et al., 2003; Liu et al., 2005; Hashim et al., 2013). The
geological map of the Dabie–Sulu orogenic belt and its corresponding lithological
distribution in the southern Dabie–Sulu region are displayed in Figure 9. As one of the
largest UHPM belts in the world for Dabie–Sulu orogen, gneiss is the outcropping rock
directly in contact with eclogite, and occupies up to 90% of the exposed metamorphic
rock area. Therefore, the in situ laboratory-based electrical conductivity of gneiss at
high temperature and pressure is very significant to interpret the conductivity structure
in the Dabie–Sulu belt, deep in the Earth's interior. The Dabie terrane is a major
segment bounded by the Tan–Lu fault to the east and separated into a series of
continuous zones by several large-scale E–W trending faults; the Sulu terrane is
segmented into a number of blocks by several NE–SW trending faults subparallel to the
Tan–Lu fault (Zheng, 2008; Xu et al., 2013). The discovery of coesite and/or diamond
inclusions in various types of rock (e.g., gneiss, eclogite, amphibolite, marble and
jadeite quartzite) through the Dabie–Sulu orogen indicates that continental crust has
been subducted at a depth of 80–200 km and subsequently exhumed to the Earth's
surface. During subduction, dehydration reactions of some hydrous minerals (e.g.,
lawsonite, phengite and chlorite) and partial melting of other regional metamorphic
rocks (e.g., gneiss and eclogite) occur at high temperature and pressure (Xu et al., 2013;
Liu et al., 2014). Previous field MT results have found that high conductivity anomalies
with magnitudes of $10^{-1}$ S/m are widely distributed at 10–20 km in the Dabie–Sulu
UHPM belt (Xiao et al., 2007). In addition, the slab-like high velocity anomaly results
have also confirmed a depth of ≥110 km for the uppermost mantle beneath the Dabie–

Sulu orogen, which represents a remnant of the subducted Yangtze block after Triassic continent–continent collision (Xu et al., 2001). However, the origin and causal mechanisms of these high conductivity anomalies for the Dabie–Sulu UHPM belt are still unknown. Together with the two main constituent rocks (natural eclogite and granulite) in the UHPM belt, it is crucial to explore whether the gneiss electrical conductivity can be used to interpret the high conductivity anomalies distributed in the Dabie–Sulu tectonic belt. The relationship between temperature and depth in the Earth's stationary crust can be obtained by a numerical solution of the heat conduction equation (Selway et al., 2014):

$$T = T_0 + (\frac{Q}{k})Z - (\frac{A_0}{2k})Z^2 \tag{4}$$

where $T_0$ is the surface temperature (K), $Q$ is the surface heat flow (mW/m$^2$), $Z$ is the lithosphere layer depth (km), $k$ is thermal conductivity (W/mK), and $A_0$ is the lithospheric radiogenic heat productivity ($\mu$W/m$^3$). Based on previous studies, the corresponding thermal calculation parameters for the Dabie–Sulu orogen are $Q$=75 mW/m$^2$ (He et al., 2009), $A_0$=0.31 $\mu$W/m$^3$ and $k$=2.6 W/mK (Zhou et al., 2011).

Based on the heat conduction equation (Eq. 4) and thermal calculation parameters, the conductivity–temperature results of gneiss with various chemical compositions ($W_A$=Na$_2$O+K$_2$O+CaO=7.12%, 7.27% and 7.64%) can be converted to a conductivity–depth profile for the Dabie–Sulu orogen (Fig. 10). A similar transformation was also conducted for granulite by Fuji-ta et al. (2004) and eclogite with different oxygen fugacity (Cu+CuO, Ni+NiO, and Mo+MoO$_2$) by Dai et al. (2016). Figure 10 makes clear that the high conductivity anomaly of 10$^{-1.5}$–10$^{-0.5}$ S/m from the field MT results in the Dabie–Sulu UHPM belt occurs at 12–21 km, compared with three dominant constituent rock conductivities of gneiss, granulite and eclogite in the region. Although our obtained electrical conductivity of gneiss with different chemical compositions is moderately higher than granulite and eclogite, it is not high enough to explain the high conductivity anomaly observed in field MT results in the Dabie–Sulu orogen. In other words, three dominant outcrops of metamorphic rocks, including gneiss, eclogite and granulite, are not substances that produce the high conductivity anomalies of the Dabie–

Sulu orogen. However, the conductivity–depth profile for gneiss with various chemical
compositions may provide an important constraint on the interpretation of field
magnetotelluric conductivity results in the regional UHPM belt.
Aside from the chemical composition, other available alternative causes for high
conductivity anomalies can be considered, such as water in nominally anhydrous
minerals (Wang et al., 2006; Yang, 2011; Dai and Karato, 2009, 2014a), interconnected
saline (or aqueous) fluids (Hashim et al., 2013; Shimojuku et al., 2014; Sinmyo and
Keppler, 2017; Guo et al., et al., 2015; Li et al., 2018), partial melting (Wei et al., 2001;
Maumus et al., 2005; Gaillard et al., 2008; Ferri et al., 2013; Laumonier et al., 2015,
2017; Ghosh and Karki, 2017), interconnected secondary high-conductivity phases (e.g.,
FeS, $Fe_3O_4$) (Jones et al., 2005; Bagdassarov et al., 2009; Padilha et al., 2015),
dehydration of hydrous minerals (Wang et al., 2012, 2017; Manthilake et al., 2015, 2016;
Hu et al., 2017; Sun et al., 2017a, b; Chen et al., 2018) and graphite films on mineral
grain boundaries (Freund, 2003; Pous et al., 2004; Chen et al., 2017). In consideration
of the similar formation conduction and geotectonic environments, the Himalaya–
Tibetan orogenic system was compared with the Dabie–Sulu UHPM belt, and explained
high electrical conductivity anomalies. Previous evidences from magnetotelluric and
elastic seismic velocity data in the southern Tibet and northwestern Himalaya zones
have confirmed that the high conductivity and low seismic velocity anomalies
widespread exist at 10–25 km in the Himalaya–Tibetan orogenic system (Wei et al.,
2001; Unsworth et al., 2005; Arora et al., 2007; Caldwell et al., 2009). Some studies
have hypothesized that partial melting is the cause of the high conductivity anomalies
in the Himalaya–Tibetan orogenic system (Wei et al., 2001; Gaillard et al., 2004;
Hashim et al., 2013). Nevertheless other researchers think they are closely related with
interconnected aqueous fluid (Makovsky and Klempere, 1999). As argued by Li et al.
(2003), five possible hypotheses could explain the cause for the high conductivity
anomalies in the INDEPTH magnetotelluric data of the southern Tibet mid-crust. The
authors found that the high conductivity anomalies may be a result of interconnected
melt and fluids. Recently, Naif et al. (2018) suggested that the high conductivity
anomaly at 50–150 km can be explained by either a small amount of water stored in

nominally anhydrous minerals or interconnected partial melts. In the present study, the electrical conductivity of gneiss with various chemical compositions at high temperature and pressure cannot be used to interpret the high conductivity anomalies of the Dabie–Sulu UHPM belt. Therefore, we propose that it is possibly caused by interconnected fluids or melts that result in high conductivity anomalies for the Dabie–Sulu UHPM belt.

**6 Conclusions**

The electrical conductivity range of gneiss samples with various chemical compositions was about $10^{-5}$–$10^{-1}$ S/m at 623–1073 K and 0.5–2.0 GPa. Electrical conductivity of the gneiss samples significantly increased with temperature, and weakly increased with pressure. The total alkaline ion content of $K_2O$, $Na_2O$ and $CaO$ is a remarkable influence on the electrical conductivity of the gneiss samples. Based on various activation enthalpy ranges (0.35–0.52 eV and 0.76–0.87 eV), corresponding to higher and lower temperature regions at 1.5 GPa, two main conduction mechanisms are suggested to dominate the conductivity of gneiss: impurity conduction in the lower temperature region and ionic conduction (charge carriers are $K^+$, $Na^+$ and $Ca^{2+}$) in the higher temperature region. Because of the much lower conductivity of gneiss samples at high temperature and pressure, we confirmed that gneiss with various chemical compositions cannot cause the high conductivity anomalies in the Dabie–Sulu UHPM belt.

*Acknowledgements.* We thank the editor of Professor Ulrike Werban and three
anonymous reviewers for their very constructive comments and suggestions in the
reviewing process, which helped us greatly in improving the manuscript. We appreciate
Dr Kara Bogus from Edanz Group (www.edanzediting.com/ac) Scientific Editing
Company for their helps in English improvements of the manuscript. This research was
financially supported by the Strategic Priority Research Program (B) of the Chinese
Academy of Sciences (XDB 18010401), Key Research Program of Frontier Sciences
of CAS (QYZDB-SSW-DQC009), "135" Program of the Institute of Geochemistry of
CAS, Hundred Talents Program of CAS and NSF of China (41474078, 41774099 and
458 41772042).

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

**Fig. 1** Photomicrographs and electron backscattered images of three natural gneiss samples under the polarizing microscope. Pl=plagioclase; Qtz = quartz and Bt = biotite.

**Fig. 2** Experimental setup for electrical conductivity measurements at high temperatures and pressures.

**Fig. 3** Representative complex impedance spectra for run DS12 gneiss under conditions of 1.5 GPa and 623–1073 K.

**Fig 4.** Real and imaginary parts of complex impedance as functions of the measured frequencies for the runs DS13 and DS14 gneiss samples under conditions of 1.5 GPa and 623–1073 K. (a) real and (b) imaginary parts for the run DS13 gneiss; (c) real and (d) imaginary parts for the run DS14 gneiss.

**Fig. 5** Logarithm of the electrical conductivities versus the reciprocal temperatures for run DS12 gneiss during two heating/cooling cycles at 1.5 GPa.

**Fig. 6** Logarithm of the electrical conductivities versus the reciprocal temperatures for run DS12 gneiss at 0.5–2.5 GPa and 623–1073 K.

**Fig. 7** Logarithm of the electrical conductivities versus the reciprocal temperatures of the gneiss samples with various chemical compositions at 1.5 GPa and 623–1073 K.

**Fig. 8** Comparisons of the electrical conductivities of the gneiss samples measured at 1.5 GPa in this study and in previous studies. The dashed blue and green lines represent the electrical conductivities of granulite and gneiss at 1.0 GPa from Fuji-ta et al. (2004) and Fuji-ta et al. (2007), respectively, the dashed orange line represents the electrical conductivity of quartz at 1.0 GPa from Wang et al. (2010), the dashed dark green line represents the electrical conductivity of alkali feldspars at 1.0 GPa from Hu et al. (2013), the dashed sky blue line represents the electrical conductivity of natural PP216 metapelite at 0.3 GPa from Hashim et al. (2013), the dashed violet line represents the electrical conductivity of the residual JOY2-X4 enclave at 0.3 GPa from Ferri et al. (2013), the dashed red lines represent the electrical conductivity of granite at 0.5 GPa from Dai et al. (2014), and the dashed

719 pink line represents the electrical conductivity of phlogopite at 1.0 GPa from Li et

al. (2016).

**Fig. 9** Geological sketch map of the Dabie-Sulu orogenic belt (a) and its correspondent
lithological distribution diagram in the southern counterpart of Dabie-Sulu region
(b) (modified after Xu et al., 2013; Liu et al., 2014).
**Fig. 10** Laboratory-based conductivity–depth profiles constructed from data of the
gneiss samples, and the thermodynamic parameters, and comparison with
geophysically inferred field results from Dabie–Sulu UHPM belt, China. The red
solid lines represent the conductivity–depth profiles based on the conductivities of
the samples described in Fig. 3 and based on a surface heat flow of 75 mW/m$^2$ in
Dabie–Sulu UHPM belt. The dashed blue lines represent the conductivity–depth
profiles based on the conductivities of eclogite, and the dashed brown line
represents the conductivity–depth profiles based on the conductivities of granulite
(Fuji-ta et al., 2004; Dai et al., 2016). The green region represents the MT data
derived from high conductivity anormaly in Dabie–Sulu UHPM belt (Xiao et al.,
2007; He et al., 2009).














**Table 1.** Mineralogical assemblage of three natural gneiss samples. Pl=plagioclase, Qz=quartz and
Bi=biotite.

| Run No. | Mineralogical associations |
|---------|----------------------------|
| DS12 | Pl (50%) + Qz (40%) + Bi (10%) |
| DS13 | Pl (25%) + Qz (40%) + Bi (35%) |
| DS14 | Pl (60%) + Qz (25%) + Bi (15%) |


**Table 2.** Chemical composition of whole rock analysis by X-ray fluorescence (XRF) for three
gneiss samples.

| Oxides (wt.%) | DS12 | DS13 | DS14 |
|---|---|---|---|
| $SiO_2$ | 64.40 | 68.59 | 69.87 |
| $Al_2O_3$ | 15.30 | 13.62 | 14.88 |
| MgO | 3.15 | 3.00 | 1.78 |
| CaO | 1.61 | 2.48 | 0.52 |
| $Na_2O$ | 2.27 | 2.46 | 2.26 |
| $K_2O$ | 3.24 | 2.33 | 4.86 |
| $Fe_2O_3$ | 6.28 | 5.57 | 3.37 |
| $TiO_2$ | 0.81 | 0.61 | 0.38 |
| $Cr_2O_3$ | 0.02 | 0.02 | 0.01 |
| MnO | 0.08 | 0.07 | 0.03 |
| BaO | 0.06 | 0.02 | 0.12 |
| SrO | 0.03 | 0.03 | 0.02 |
| $P_2O_5$ | 0.19 | 0.16 | 0.08 |
| $SO_3$ | <0.01 | <0.01 | 0.28 |
| L.O.I | 1.89 | 0.86 | 1.67 |
| Total | 99.33 | 99.82 | 100.13 |


**Table 3.** Fitted parameters of the Arrhenius relation for the electrical conductivity of three gneiss
samples. Two equations of $\sigma = \sigma_0 \exp(-\Delta H / kT)$ and $\Delta H = \Delta U + P\Delta V$ are adopted, in here, $\sigma_0$
is the pre-exponential factor (S/m), $\Delta H$ is the activation enthalpy (eV), $k$ is the Boltzmann constant
(eV/K), $T$ is the absolute temperature (K), $\Delta U$ is the activation energy (eV), $P$ is the pressure (GPa)
and $\Delta V$ is the activation volume (cm³/mole). The symbol of $R^2$ is denoted as the fitted correlation
coefficient.

| Run No. | $P$ (GPa) | $T$ (K) | Log $\sigma_0$ (S/m) | $\Delta H$ (eV) | $R^2$ | $\Delta U$ (eV) | $\Delta V$ (cm³/mole) |
|---|---|---|---|---|---|---|---|
| | 0.5 | 623–723 | -0.20±0.09 | 0.58±0.01 | 99.91 | | |
| | 1.0 | 623–723 | -0.06±0.01 | 0.56±0.01 | 99.99 | 0.63±0.01 | -7.10±0.92 |
| | 1.5 | 623–773 | -0.06±0.02 | 0.52±0.02 | 99.66 | | |
| | 2.0 | 623–773 | -0.38±0.05 | 0.47±0.01 | 99.96 | | |
| DS12 | 0.5 | 773–1073 | 1.11±0.08 | 0.77±0.01 | 99.79 | | |
| | 1.0 | 773–1073 | 0.98±0.08 | 0.72±0.01 | 99.77 | 0.78±0.03 | -2.69±2.40 |
| | 1.5 | 823–1073 | 1.43±0.05 | 0.76±0.01 | 99.93 | | |
| | 2.0 | 823–1073 | 1.26±0.11 | 0.71±0.03 | 99.51 | | |
| DS13 | 1.5 | 623–773 | -0.92±0.04 | 0.35±0.01 | 99.93 | | |
| | | 823–1073 | 2.26±0.12 | 0.84±0.01 | 99.66 | / | / |
| DS14 | 1.5 | 623–773 | -0.49±0.10 | 0.38±0.01 | 99.60 | | |
| | | 823–1073 | 2.63±0.10 | 0.87±0.02 | 99.81 | | |










**Figure 1**

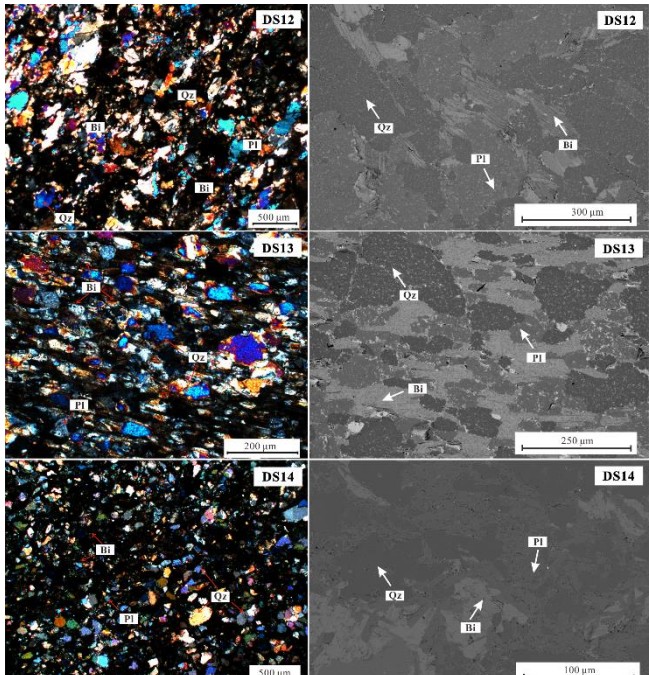



















**Figure 2**

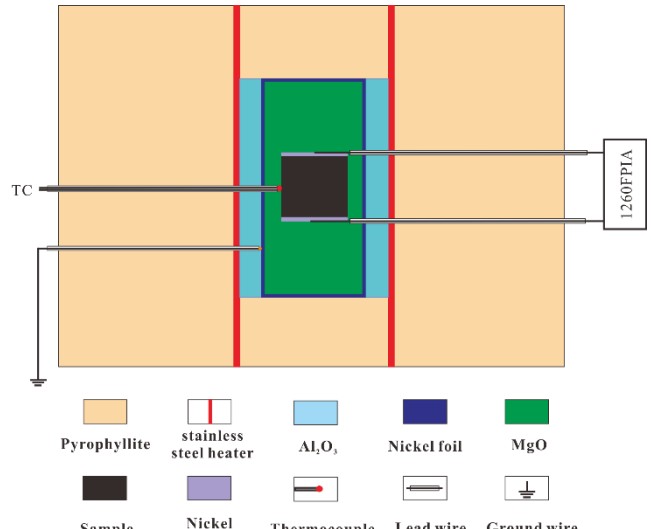





















**Figure 3**

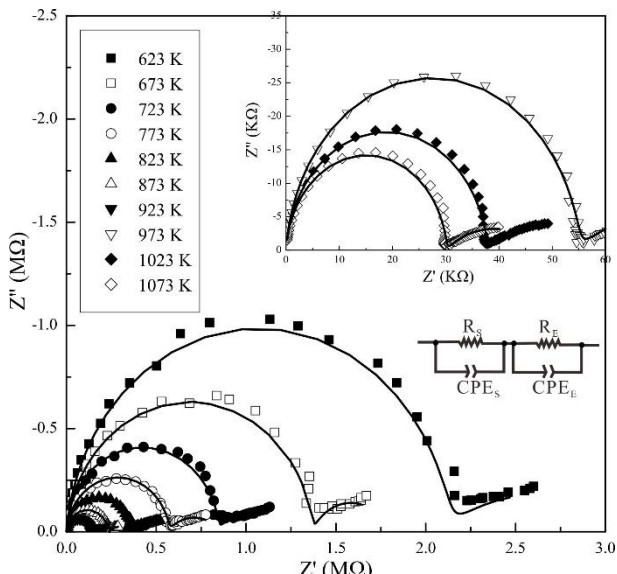





















**Figure 4**

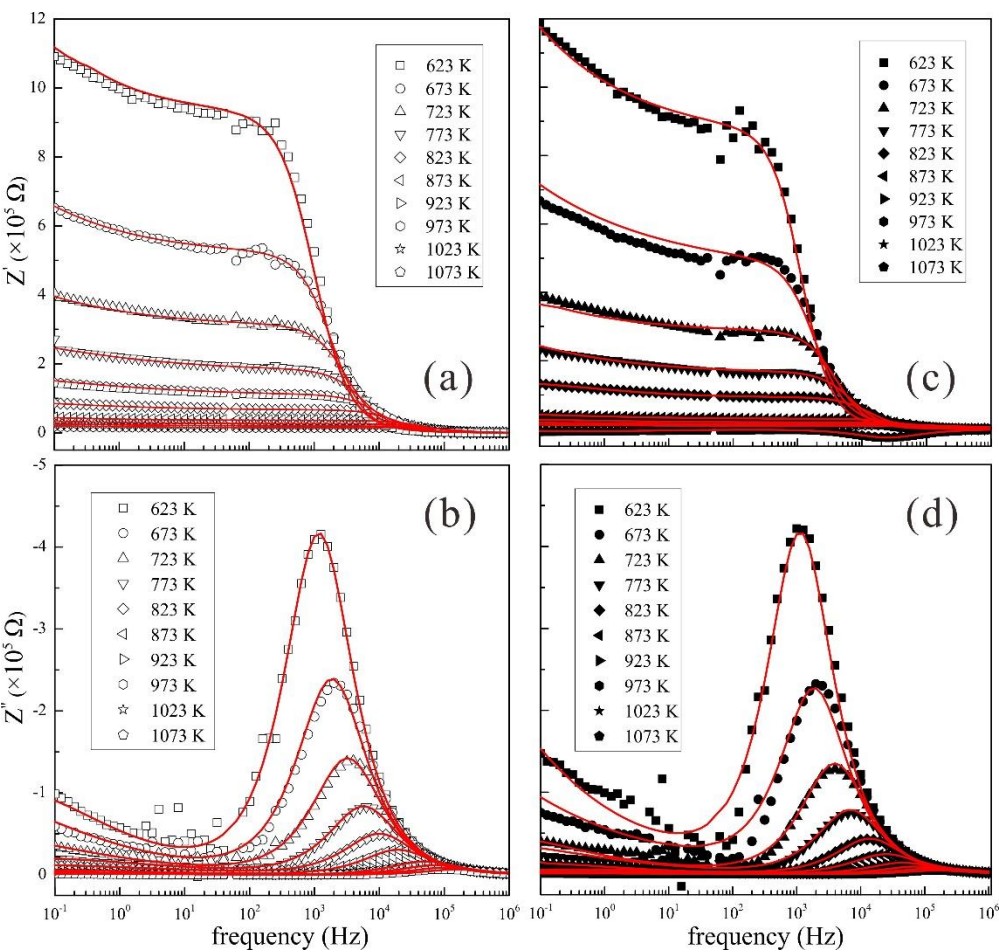

**Figure 5**

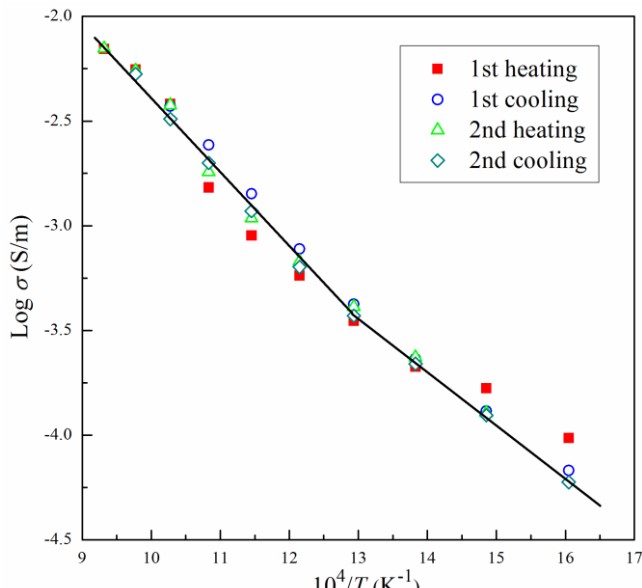





















**Figure 6**

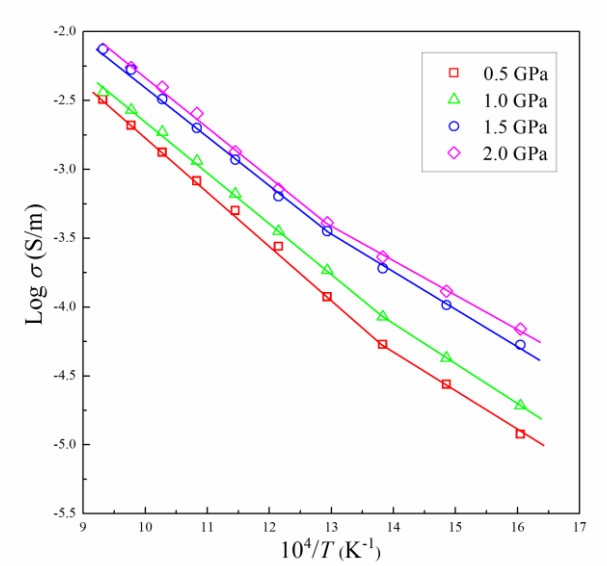





















**Figure 7**

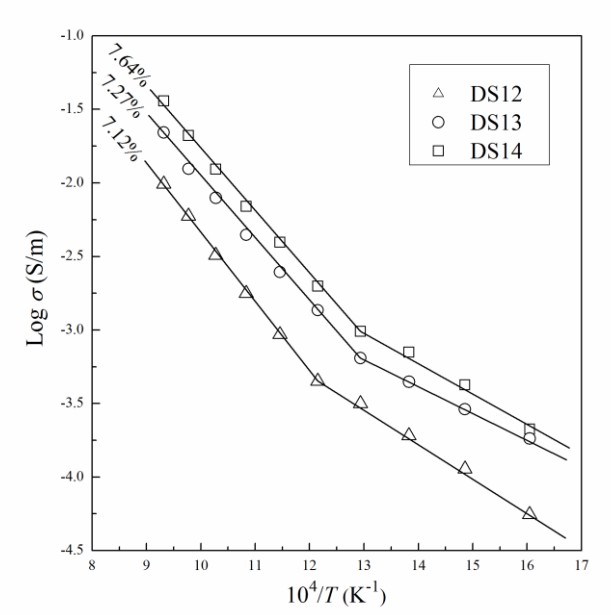


**Figure 8**

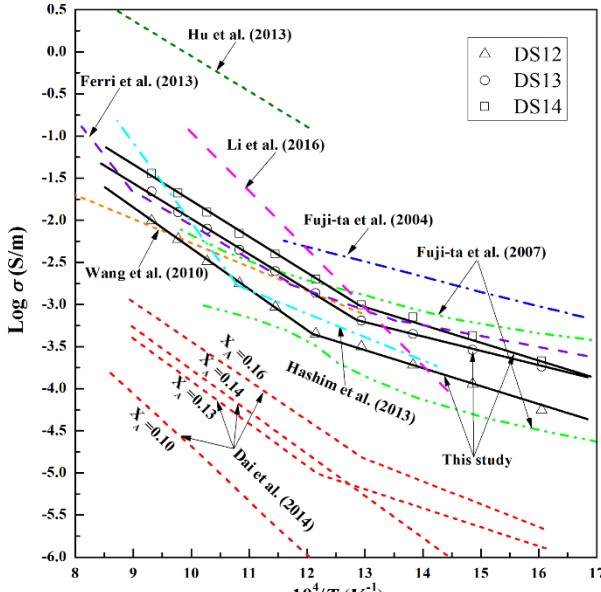


**Figure 9**

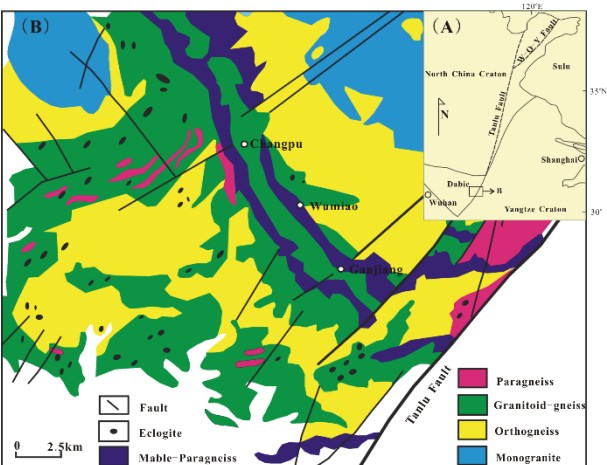























**Figure 10**

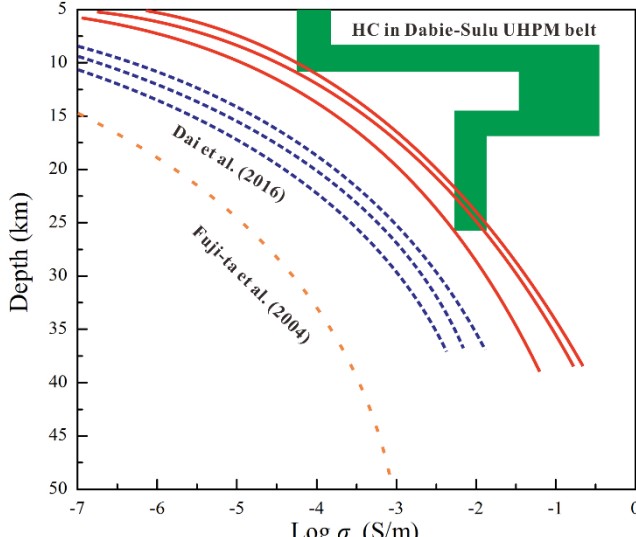
