# Peer review of "Effect of chemical composition on the electrical conductivity of # gneiss at high temperatures and pressures"

_Solid Earth, 2017_

## Referee Comment (RC1) · Anonymous Referee #1 · 12 Oct 2017

This article reports the effect of chemical composition on the electrical conductivity of biotite-bearing felsic gneiss at high P-T conditions. They tried to explain the conductivity differences by the contribution of total $K^+ + Na^+ + Ca^{2+}$ of three natural gneiss samples. The experimental technique is top-notch but the strategy and discussion are not convincing. I think the manuscript must be revised largely and more evidences should be provided before publication.

The authors measured the electrical conductivity of gneiss parallel to foliation. There are at least two reasons may contribute to the conductivity differences, including chemical composition effect and textural difference. How to evaluate the effect of textures?

[Figure]

Biotite usually deforms and aggregates to form the band texture and it may exhibit strong conductivity anisotropy, highest along the layered surface and lowest perpendicular to the layered structure. The conductivity differences, therefore, may result from the texture differences. The authors did not describe the samples carefully. Even that the effect of chemical compositions dominates on the conductivities, the authors cannot use the composition data of a whole rock as that of the unique sample used in conductivity measurement because of the inhomogeneity. To overcome these uncertainties, well mixed powder samples must be used instead although the geological application will be penalised. It is also a strange strategy that the authors havn't choosed the samples from Dabie-Sulu as the starting materials, despite finally they apply the results to explian the HCL within Dabie-Sulu.

Other comments: (1) Quality of writing: In its present state, this article is not publishable. Writing needs tremendous improvements to match the requirements of any peer-reviewed journal. (2) The authors should calculate the activation volume for Run DS12, and explain the possible mechanism of positive pressure effect on the conductivity. (3) Line 322-325: The authors should clearly show how to convert the conductivity-temperature data to conductivity-depth profile with the aid of heat flow for the general readers.

---

## Referee Comment (RC2) · Anonymous Referee #2 · 23 Oct 2017

In their submitted manuscript the authors investigate the electrical properties of different gneiss samples at elevated temperatures and high hydrostatic pressures by means of state of the art experimental facilities. The paper focuses on the effect of the chemical composition to the measured conductivity and different conduction mechanisms are reported. Geophysical implication is also discussed. The work is interesting and worth publishing but additional aspects could also be revealed after further analysis of the experimental data. The authors should pay much effort to improve the quality of their work, in order to be suitable for publication. The following issues should be carefully addressed: 1. In my opinion, the author should not just limited to the calculations of the dc-conductivity but also explore the advantages of the complex impedance spec-

troscopy. Otherwise, they could measure the dc-conductivity by varying linearly the temperature at different selected pressures. I suggest using also other formalisms of impedance data, such as ac-conductivity and complex impedance presentation of their data. 2. According to my previous comment, it would be also desirable to present the results of all the measured samples (or at least of 2 of them) in suitable figures, i.e. real and imaginary parts of ac-conductivity and impedance as a function of the measured frequency at different T and P, except of the Cole-Cole plots of complex impedance. 3. In the measured frequency range (0.1 Hz-1 MHz) the overall conductivity should usually include contributions from grains interior, grain boundaries and electrodes polarization. In their fitting procedure the authors included only two types of contributions, with the main one the bulk conductivity. It has to be clarified if this refers to both grains interior and grain boundaries or only to the conductivity of the grains interior. In the former case, the 2 contributions should be separated. 4. An important finding which should be emphasized because it is rarely observed in minerals and rocks is the negative activation volumes that are observed, i.e. increase of conductivity with pressure. Their values should be calculated and compared with the activation volumes of the constituent minerals (biotite, feldspar and quartz) and/or other possible reported values of gneiss. Possible reasons for this finding should be also discussed. In fact, it is the effective activation volume that is found to have negative values and could be related to the influence of percolation effects in the grain boundaries. 5. Lines 208-211, "... the gneiss samples were unstable in the first heating cycle." This could arise from the existence of bound water that is trapped in grain boundaries or in the rock structure in the form of hydroxyls and is desorbed at high temperatures. In this sense, the conduction mechanism of low activation energies at the low temperature region could be related to proton conduction. The corresponding ac-conductivity spectra might give insights to these issues. This alternative explanation should be checked. Furthermore, the manuscript should be carefully revised to improve the quality of the English language. Some less important issues that have to be addressed: 6. Line 73: for the sake of completeness it would be desirable to briefly refer to these different types of

gneisses. 7. Lines 96, 102, 106, 493: the measured specimens are 3, not 4, as stated incorrectly. 8. Lines 155-156: It would better to use the symbol CPE for the constant phase element, instead of Cs which corresponds to a capacitor. 9. Table 3: I suppose that the last column corresponds to the correlation coefficients of the fitting procedure. Please change the symbol (greek gamma) to the correct one, R. In addition, taking into account the constructive comments of the 1st referee, I would suggest that the paper could focus not only to the effect of the chemical composition to the measured conductivity but also to the negative values of activation volumes, the geophysical implication that already exists in the manuscript and to the detailed investigation of the complex impedance spectra. In this sense, the title could be more general without focusing to the influence of chemical composition on the measured conductivity. For example "Complex impedance spectroscopy of gneiss samples at high temperatures and pressures".
* * *

---

## Author Comment (AC1) · 6 Nov 2017

Anonymous Referee 1: This article reports the effect of chemical composition on the electrical conductivity of biotite-bearing felsic gneiss at high P-T conditions. They tried to explain the conductivity differences by the contribution of total $K^+ + Na^+ + Ca^{2+}$ of three natural gneiss samples. The experimental technique is top-notch but the strategy and discussion are not convincing. Thanks for the positive comments. In this revised manuscript, we conscientiously read through all valuable comments and suggestions, and revised each

one points by points, sentences by sentences. So far we have made some substantial strategy and discussion convinced in the revised manuscript.

1. I think the manuscript must be revised largely and more evidences should be provided before publication. The authors measured the electrical conductivity of gneiss parallel to foliation. There are at least two reasons may contribute to the conductivity differences, including chemical composition effect and textural difference. How to evaluate the effect of textures? Biotite usually deforms and aggregates to form the band texture and it may exhibit strong conductivity anisotropy, highest along the layered surface and lowest perpendicular to the layered structure. The conductivity differences, therefore, may result from the texture differences. The authors did not describe the samples carefully. Thanks for your valuable and professional comments and suggestions. Indeed, just as described by the first anonymous reviewer, it is possibly existing two dominant reasons of chemical composition and texture that can result in the difference of electrical conductivity measurement results. Based on the results of the previously reported studies, the main conduction mechanism for phlogopite is ionic conduction, and $K+$ is proposed to be the main charge carriers (Li et al. 2017a, b). We suggested that the charge carriers of the gneiss samples were $K+$, $Na+$ and $Ca2+$. Therefore, the influence of biotite on the conductivities of gneiss has been taken into consideration. On the other hand, the electrical conductivities of the gneiss samples don't regularly increase with increasing content of biotite, as shown in Table 1 and Fig. 6. Based on all of these obtained experimental results, it made clear that the content of biotite is not the main influence factor influence on the electrical conductivity of gneiss samples. In the present studies, we considered the gneiss sample as a whole to explore its electrical conductivity at high temperature and high pressure, and it is crucial that the chemical composition of sample (WA = Na2O+K2O+CaO = 7.12On the base of the valuable suggestion from the anonymous reviewer, we have already supplemented a large quantity of detailed description in the section of 2.1 sample preparation in the revised manuscript. Some main revisions have been summarized as follows: Three relatively homogeneous natural gneiss samples with a parallel to foliation direction

were collected from Xinjiang, China. The surface of the sample is fresh, non-fractured and non-oxidized, without evidence of alteration before and after experiments. The main rock-forming minerals of three gneiss samples are feldspar, quartz and biotite, respectively. It was indicated that three gneiss samples have the same mineralogical assemblage, and all of them belong to biotite-bearing felsic gneiss. From Table 2, we found that the totally alkali- (such as $K^+$ and $Na^+$) and alkali-Earth ($Ca^{2+}$) metallic ion content for each sample were various. And therefore, in the present studies, we have conducted a series of experiments in order to determine the influence of chemical composition by changing the totally alkali- and alkali-Earth metallic ion content on the electrical conductivity of gneiss at high temperature and high pressure.

2. Even that the effect of chemical compositions dominates on the conductivities, the authors cannot use the composition data of a whole rock as that of the unique sample used in conductivity measurement because of the inhomogeneity. To overcome these uncertainties, well mixed powder samples must be used instead although the geological application will be penalized. Thanks for your professional comments and advisements. Indeed, it is one inevitable problem of the sample's inhomogeneity only if the researcher tried to measure the electrical conductivity of natural rock at high temperature and high pressure. Just as described by the anonymous reviewer, it's true that chemical composition for hot-pressed sintering sample by the mixed powder samples seems much more homogeneous than those of natural samples. In this study, we chose a series of natural samples rather than hot-pressed sintering sample, mainly considered: (a) the structure of mixed powder sample is completely different from that of natural sample, which implies that the natural sample become more representative to explore its geophysical implications; (b) In the process of hot-pressed sintering sample, grain size is difficult to control for each experiment, and therefore, the grain size influence on the electrical conductivity issue for one complex rock is not easy to be well solved; (c) Only if one natural rock sample of its mineralogical assembly contained one or several hydrous minerals, such as amphibole, mica et al., it is not strongly suggested that we chose one hot-pressed sintering method to synthesize the starting experimental sample. If the hot-pressed temperature is too low, I am afraid that some inevitable fractures and microcrackings have some influences on the subsequent electrical conductivity measurement. On the contrary, if the hot-pressed temperature is too high, the dehydration of hydrous mineral must be full considered in the process of sample preparation. As a matter of fact, in our previously reported papers, we have already completed electrical conductivity measurements on many representative natural rock samples at high temperature and high pressure in our laboratory, such as natural samples: pyroxenite (Dai et al. 2006), lherzolite (Dai et al. 2008), amphibolite (Zhou et al. 2011; Wang et al. 2012), granite (Dai et al. 2014), basalt (Dai et al. 2015), gabbro (Dai et al. 2015), and eclogite (Dai et al. 2016), etc. In addition, much more papers on the electrical conductivity of natural rocks have been also published in other laboratory, such as granulite (Fuji-ta et al. 2004), gneiss (Fuji-ta et al. 2007), amphibolite (Saltas et al. 2013), and quartzite (Shimojuku et al. 2014), etc. In addition, we made great efforts in choosing small area of three relatively homogeneous natural gneiss samples with a parallel to foliation direction in the process of our current sample preparation. During the conductivity measurements, we cut and polish them into a cylinder of $\Phi$ 6.0 $\times$ 6.0 mm in order to efficiently avoid this issue. Of course, in the future, we can try to measure one hot-pressed synthetic gneiss sample and compare it.

3. It is also a strange strategy that the authors haven't choose the samples from Dabie-Sulu as the starting materials, despite finally they apply the results to explain the HCL within Dabie-Sulu. Thanks for your valuable comments. To be frank, due to some practical difficulties for our own work area, we didn't collect a series of natural gneiss samples originated from the region of Dabie-Sulu ultrahigh-pressure metamorphic belt. However, it has been confirmed that abundant felsic gneisses were widespread distributed in Dabie-Sulu ultrahigh-pressure metamorphic belt, and the mineralogical assemblage of gneiss in Dabie-Sulu ultrahigh-pressure metamorphic belt is similar to that of our present experimental samples (Gong et al. 2013). In addition, the gneiss distributed in the deep Earth interior may be existing some discrepancy from that of outcrop in the Earth's surface. Three gneisses with various chemical compositions are

able to represent many natural biotite-bearing felsic gneiss, and we arrived in one conclusion that the electrical conductivities of gneiss cannot be used to interpret the high conductivity layers (HCLs) in Dabie-Sulu ultrahigh-pressure metamorphic belt.

Other comments: (1) Quality of writing: In its present state, this article is not publishable. Writing needs tremendous improvements to match the requirements of any peer-reviewed journal. As for the issue of English language, we appreciated Dr Aaron Stallard in Stallard Scientific Editing Company for their helps in English improvements of the manuscript. The substantial corrections for English have been conducted sentences by sentences. After that, the revised paper becomes much more easily be read and understood.

(2) The authors should calculate the activation volume for Run DS12, and explain the possible mechanism of positive pressure effect on the conductivity. According to the suggestion, we have already supplemented all of these results on the activation volume for Run DS12 and the calculating equation. With increasing pressure, the electrical conductivity of gneiss increases, accordingly. The activation volumes for Run DS12 are -7.10 cm3/mole and -2.69 cm3/mole at low temperature region and high temperature region, respectively. Another one representative metamorphic rock for gneiss, we can compared it with the electrical conductivity of eclogite. Recently, Dai et al. (2016) measured the electrical conductivity of dry eclogite, and the obtained negative activation volume value for eclogite is -2.51 cm3/mole under conditions of 1.0-3.0 GPa and 873-1173 K. It was proposed that the main conduction mechanism for dry eclogite is intrinsic conduction (Dai et al. 2016). The conduction mechanism for gneiss sample at high temperature region was also proposed to be intrinsic conduction, but the conduction mechanism at low temperature region was impurity conduction (possible charge carriers: K+, Na+, Ca2+, H+, et al.). In addition, it was suggested that the positive pressure effect on the electrical conductivities of gneiss samples may be due to the more complicated rock structure.

(3) Line 322-325: The authors should clearly show how to convert the conductivity

temperature data to conductivity-depth profile with the aid of heat flow for the general readers. Thanks for your professional and precious suggestions. The relationship between temperature and depth in the Earth's stationary crust can be described by a numerical solution of the heat conduction equation (Čermák and Laštovičková 1987): (1) where $T_0$ is the surface temperature (K), Q is the surface heat flow (mW/m2), Z is the lithospheric layer depth (km), k is thermal conductivity (W/mK), and $A_0$ is the lithospheric radiogenic heat productivity ($\mu$W/m3). Based on previous studies, the thermal calculation parameters for Dabie-Sulu terrane are Q = 75 mW/m2 (He et al. 2009), $A_0$ = 0.31 $\mu$W/m3, and k = 2.6 W/mK (Zhou et al. 2011). According to heat conduction equation and thermal calculation parameters, conductivity-temperature data can be converted to conductivity-depth profile for Dabie-Sulu terrane.

References Čermák, V. and Laštovičková, M.: Temperature profiles in the earth of importance to deep electrical conductivity models. Pure Appl. Geophys., 25, 255‒284, 1987. Dai, L.D., Hu, H.Y., Li, H.P., Wu, L., Hui, K.S., Jiang, J.J., and Sun, W.Q.: Influence of temperature, pressure, and oxygen fugacity on the electrical conductivity of dry eclogite, and geophysical implications. Geochem. Geophys. Geosyst., 17, 2394–2407, 2016. Dai, L.D., Hu, H.Y., Li, H.P., Hui, K.S., Jiang, J.J., Li, J., and Sun, W.Q.: Electrical conductivity of gabbro: the effects of temperature, pressure and oxygen fugacity. Eur. J. Mineral., 27, 215–224, 2015. Dai, L.D., Jiang, J.J., Li, H.P., Hu, H.Y., and Hui, K.S.: Electrical conductivity of hydrous natural basalts at high temperatures and pressures. J. Appl. Geophys., 112, 290‒297, 2015. Dai, L.D., Hu, H.Y., Li, H.P., Jiang, J.J., and Hui, K.S.: Influence of temperature, pressure, and chemical composition on the electrical conductivity of granite. Am. Mineral., 99, 1420‒1428, 2014. Dai, L.D., Li, H.P., Deng, H.M., Liu, C.Q., Su, G.L., Shan, S.M., Zhang, L., and Wang, R.P.: In-situ control of different oxygen fugacity experimental study on the electrical conductivity of lherzolite at high temperature and high pressure. J. Phys. Chem. Solids, 69, 101‒110, 2008. Dai, L.D., Li, H.P., Liu, C.Q., Su, G.L., and Shan, S.M.: Experimental measurement of the electrical conductivity of pyroxenite at high temperature and high pressure under different oxygen fugacities. High

Pressure Res., 26, 193–202, 2006. Fuji-ta, K., Katsura, T., Matsuzaki, T., Ichiki, M., and Kobayashi, T.: Electrical conductivity measurement of gneiss under mid- to lower crustal P-T conditions. Tectonophysics, 434, 93–101, 2007. Fuji-ta, K., Katsura, T., and Tainosho, Y.: Electrical conductivity measurement of granulite under mid- to lower crustal pressure-temperature conditions. Geophys. J. Int., 157, 79–86, 2004. Gong, B., Chen, R.X., and Zheng, Y.F.: Water contents and hydrogen isotopes in nominally anhydrous minerals from UHP metamorphic rocks in the Dabie-Sulu orogenic belt. Chinese Sci. Bull., 58, 4384–4389, 2013. He, L., Hu, S., Yang, W., and Wang, J.: Radiogenic heat production in the lithosphere of Sulu ultrahigh-pressure metamorphic belt. Earth Planet. Sci. Lett., 277, 525–538, 2009. Li, Y., Yang, X.Z., Yu, J.H., and Cai, Y.F.: Unusually high electrical conductivity of phlogopite: the possible role of fluorine and geophysical implications. Contrib. Mineral. Petrol., 171, 37, 2016. Li, Y., Jiang, H.T., and Yang X.Z.: Fluorine follows water: Effect on electrical conductivity of silicate minerals by experimental constraints from phlogopite. Geochim. Cosmochim. Ac., 217, 16–27, 2017. Saltas, V., Chatzistamou, V., Pentari, D., Paris, E., Triantis, D., Fitilis, I., and Vallianatos, F.: Complex electrical conductivity measurements of a KTB amphibolite sample at elevated temperatures. Mater. Chem. Phys., 139, 169‒175, 2013. Shimojuku, A., Yoshino, T., and Yamazaki, D.: Electrical conductivity of brine-bearing quartzite at 1 GPa: Implications for fluid content and salinity of the crust. Earth Planets Space, 66, 1–9, 2014. Wang, D.J., Guo, X.Y., Yu, Y.J., and Karato, S.: Electrical conductivity of amphibole-bearing rocks: influence of dehydration. Contrib. Mineral. Petrol., 164, 17‒25, 2012. Zhou, W.G., Fan, D.W., Liu, Y.G., and Xie, H.S.: Measurements of wave velocity and electrical conductivity of an amphibolite from southwestern margin of the Tarim Basin at pressures to 1.0 GPa and temperatures to 700 °C: comparison with field observations. Geophys. J. Int., 187, 1393–1404, 2011.

Please also note the supplement to this comment:
https://www.solid-earth-discuss.net/se-2017-103/se-2017-103-AC1-supplement.pdf

---

## Author Comment (AC2) · 6 Nov 2017

Anonymous Referee 2: In their submitted manuscript the authors investigate the electrical properties of different gneiss samples at elevated temperatures and high hydrostatic pressures by means of state of the art experimental facilities. The paper focuses on the effect of the chemical composition to the measured conductivity and different conduction mechanisms are reported. Geophysical implication is also discussed. The work is interesting and worth publishing but additional aspects could

also be revealed after further analysis of the experimental data. The authors should pay much effort to improve the quality of their work, in order to be suitable for publication. The following issues should be carefully addressed: We thank the anonymous reviewer for very constructive and enlightened comments and suggestions in the reviewing process, which helped us greatly in improving the manuscript. In this revised paper, we conscientiously read through all comments from the valuable suggestions of the reviewer, and revised each one points by points, sentences by sentences. All of detailed revisions and responses are listed as follows.

1. In my opinion, the author should not just limited to the calculations of the dc-conductivity but also explore the advantages of the complex impedance spectroscopy. Otherwise, they could measure the dc-conductivity by varying linearly the temperature at different selected pressures. I suggest using also other formalisms of impedance data, such as ac-conductivity and complex impedance presentation of their data. Thanks for your valuable and professional comments and suggestions. As a matter of fact, it is indeed one good idea that we can calculate the complex impedance presentation and ac-conductivity from the complex impedance spectroscopy. In the revised manuscript, we have already supplemented another one Figure 4: Real and imaginary parts of complex impedance as functions of the measured frequencies for run DS13 and DS14 gneiss under conditions of 1.5 GPa and 623–1073 K. (a) real and (b) imaginary parts for run DS13 gneiss; (c) real and (d) imaginary parts for run DS14 gneiss. In order to explore the geophysical implication from the electrical conductivities of gneiss samples, we calculated the dc-conductivities of natural gneiss samples, and researched the influence of chemical compositions, temperatures and pressures. Indeed, most previous studies calculated dc-conductivities of minerals and rocks to compare with Magnetotelluric (MT) and geomagnetic depth sounding (GDS) results (Fuji-ta et al. 2007; Dai et al. 2016; Hu et al. 2017).

2. According to my previous comment, it would be also desirable to present the results of all the measured samples (or at least of 2 of them) in suitable figures, i.e. real

and imaginary parts of ac-conductivity and impedance as a function of the measured frequency at different T and P, except of the Cole-Cole plots of complex impedance. Thanks for your valuable and professional comments and suggestions. We added the diagram about the relationship between frequency and real and imaginary parts of impedance. From the real and imaginary parts of complex impedance as functions of the measured frequencies (Figure 4), the values of real parts almost keep unchanged in the frequency of 106-104 Hz, and sharply increased in the frequency of 104-102 Hz, and slowly then increased in the low frequency region; the values of imaginary slowly increased in the frequency of 106-105 Hz, and sharply increased and sharply decreased in the frequency of 105-102 Hz, and then slowly increased in the low frequency region.

Figure 4. Real and imaginary parts of complex impedance as functions of the measured frequencies for run DS13 and DS14 gneiss under conditions of 1.5 GPa and 623–1073 K. (a) real and (b) imaginary parts for run DS13 gneiss; (c) real and (d) imaginary parts for run DS14 gneiss.

3. In the measured frequency range (0.1 Hz-1 MHz) the overall conductivity should usually include contributions from grains interior, grain boundaries and electrodes polarization. In their fitting procedure the authors included only two types of contributions, with the main one the bulk conductivity. It has to be clarified if this refers to both grains interior and grain boundaries or only to the conductivity of the grains interior. In the former case, the 2 contributions should be separated. Thanks for your valuable comments. All the impedance spectra at the different temperatures contained almost ideal semicircles in the high-frequency domain and additional tails in the low-frequency domain. The ideal semicircles represent the bulk electrical properties of the sample, and the additional tails are the typical characteristic of the sample–electrode interface in diffusion processes (Roberts and Tyburczy 1991; Dai et al. 2014; Hu et al. 2015). Therefore, the bulk sample resistance can be determined by fitting the high-frequency semicircular arc. The equivalent circuit is composed of the series connection of RS–

CPES (RS and CPES represent the resistance and constant-phase element of a sample, respectively) and RE–CPEE (RE and CPEE represent the interaction of the charge carrier with the electrode). 4. An important finding which should be emphasized because it is rarely observed in minerals and rocks is the negative activation volumes that are observed, i.e. increase of conductivity with pressure. Their values should be calculated and compared with the activation volumes of the constituent minerals (biotite, feldspar and quartz) and/or other possible reported values of gneiss. Possible reasons for this finding should be also discussed. In fact, it is the effective activation volume that is found to have negative values and could be related to the influence of percolation effects in the grain boundaries. According to the suggestion, we have already supplemented all of these results on the activation volume for Run DS12 gneiss. With increasing pressure, the electrical conductivity of gneiss increases, accordingly. The activation volumes for Run DS12 gneiss are -7.10 cm3/mole and -2.69 cm3/mole at low temperature region and high temperature region, respectively. Another one representative metamorphic rock for gneiss, we can compared it with the electrical conductivity of eclogite. Recently, Dai et al. (2016) measured the electrical conductivity of dry eclogite, and the obtained negative activation volume value for eclogite is -2.51 cm3/mole under conditions of 1.0-3.0 GPa and 873-1173 K. It was proposed that the main conduction mechanism for dry eclogite is intrinsic conduction (Dai et al. 2016). The conduction mechanism for gneiss sample at high temperature region was also proposed to be intrinsic conduction, but the conduction mechanism at low temperature region was impurity conduction (possible charge carriers: K+, Na+, Ca2+, H+, et al.). In addition, it was suggested that the positive pressure effect on the electrical conductivities of gneiss samples may be due to the more complicated rock structure.

5. Lines 208-211, "... the gneiss samples were unstable in the first heating cycle." This could arise from the existence of bound water that is trapped in grain boundaries or in the rock structure in the form of hydroxyls and is desorbed at high temperatures. In this sense, the conduction mechanism of low activation energies at the low temperature region could be related to proton conduction. The corresponding ac-conductivity spectra

might give insights to these issues. This alternative explanation should be checked. Thanks for your valuable comments and suggestions. According to previous studies, the electrical conductivities of most minerals and rocks with various conduction mechanisms were unstable at the first heating cycle (Fuji-ta et al. 2004, 2007; Dai et al. 2014). We determined the activation mechanism for gneiss samples by activation enthalpies. The activation enthalpies for the gneiss samples are 0.35‒0.58 eV at lower temperature range, Dai et al. (2014) measured the electrical conductivities of granite which has the same mineralogical assemblage with gneiss samples. It was proposed that the conduction mechanism at low temperatures was impurity conduction owing to low activation enthalpy (0.5 eV). We suggested that H+ may be also one kind of charge carriers of gneiss at low temperature region, other charge carriers were proposed to be K+, Na+, Ca2+, et al.

Furthermore, the manuscript should be carefully revised to improve the quality of the English language. As for the issue of English language, we appreciated Dr Aaron Stallard in Stallard Scientific Editing Company for their helps in English improvements of the manuscript. The substantial corrections for English have been conducted sentences by sentences. After that, the revised paper becomes much more easily be read and understood.

Some less important issues that have to be addressed: 6. Line 73: for the sake of completeness it would be desirable to briefly refer to these different types of gneisses. Thanks for your valuable comments and suggestions. We have already changed this sentence Line 73: "In light of mineralogical assembly of rock-bearing dominant mineral, it is general that gneiss can be divided into plagioclase gneiss, quartz gneiss, biotite, etc." In the present studies, the rock-forming minerals of our three gneiss samples are feldspar, quartz and biotite, and the volume percentage for each correspondent rock-forming mineral in different gneiss samples were various (Table 1). It was indicated that three gneiss samples have the same mineralogical assemblage, and all of them belong to the biotite-bearing felsic gneiss.

7. Lines 96, 102, 106, 493: the measured specimens are 3, not 4, as stated incorrectly. Thanks for your conscientious comments. We have already corrected them in the revised manuscript.

8. Lines 155-156: It would better to use the symbol CPE for the constant phase element, instead of Cs which corresponds to a capacitor. Thanks for your important suggestion. The symbol CPE was used in our equivalent circuit to obtain the resistance, and the electrical conductivities of gneiss samples had little change. We have changed Cs into CPE in the revised manuscript.

9. Table 3: I suppose that the last column corresponds to the correlation coefficients of the fitting procedure. Please change the symbol (greek gamma) to the correct one, R. Thanks for your valuable comments and suggestions. We have changed the symbol (greek gamma) to the correct one, $\gamma$.

In addition, taking into account the constructive comments of the 1st referee, I would suggest that the paper could focus not only to the effect of the chemical composition to the measured conductivity but also to the negative values of activation volumes, the geophysical implication that already exists in the manuscript and to the detailed investigation of the complex impedance spectra. In this sense, the title could be more general without focusing to the influence of chemical composition on the measured conductivity. For example "Complex impedance spectroscopy of gneiss samples at high temperatures and pressures". Thanks for your valuable comments and suggestions. Indeed, it is more appropriate that the manuscript title "Complex impedance spectroscopy of gneiss samples at high temperatures and pressures". I am very appreciated that you put forward such a large quantity of enlightened and precious comments and suggestions, which helped us greatly in improving the manuscript.

References Dai, L.D., Hu, H.Y., Li, H.P., Wu, L., Hui, K.S., Jiang, J.J., and Sun, W.Q.: Influence of temperature, pressure, and oxygen fugacity on the electrical conductivity of dry eclogite, and geophysical implications. Geochem. Geophys. Geosyst., 17,

2394–2407, 2016. Dai, L.D., Hu, H.Y., Li, H.P., Jiang, J.J., and Hui, K.S.: Influence of temperature, pressure, and chemical composition on the electrical conductivity of granite. Am. Mineral., 99, 1420–1428, 2014. Fuji-ta, K., Katsura, T., Matsuzaki, T., Ichiki, M., and Kobayashi, T.: Electrical conductivity measurement of gneiss under mid- to lower crustal P-T conditions. Tectonophysics, 434, 93–101, 2007. Fuji-ta, K., Katsura, T., and Tainosho, Y.: Electrical conductivity measurement of granulite under mid- to lower crustal pressure-temperature conditions. Geophys. J. Int., 157, 79–86, 2004. Hu, H.Y., Dai, L.D., Li, H.P., Hui, K.S., and Sun, W.Q.: Influence of dehydration on the electrical conductivity of epidote and implications for high conductivity anomalies in subduction zones. J. Geophys. Res., 122, 2751–2762, 2017. Hu, H.Y., Dai, L.D., Li, H.P., Hui, K.S., and Li, J.: Temperature and pressure dependence of electrical conductivity in synthetic anorthite. Solid State Ionics, 276, 136-141, 2015. Roberts, J.J. and Tyburczy, J.A.: Frequency dependent electrical properties of polycrystalline olivine compacts. J. Geophys. Res., 96, 16205–16222, 1991.

Please also note the supplement to this comment:
https://www.solid-earth-discuss.net/se-2017-103/se-2017-103-AC2-supplement.pdf

[Figure]

[Figure]

**Fig. 1.**

Plot (a): Z' (×10⁵ Ω) vs frequency (Hz), with legend 623 K, 673 K, 723 K, 773 K, 823 K, 873 K, 923 K, 973 K, 1023 K, 1073 K

Plot (b): Z'' (×10⁵ Ω) vs frequency (Hz), with same legend

Plot (c): Z' (×10⁵ Ω) vs frequency (Hz), with filled-symbol legend 623 K–1073 K

Plot (d): Z'' (×10⁵ Ω) vs frequency (Hz), with filled-symbol legend 623 K–1073 K

**Fig. 2.**

---

## Referee Comment (RC3) · Anonymous Referee #1 · 14 Nov 2017

This is a much improved submission that most questions have been well answered. I would just want to know how to exclude the effect of iron content on the bulk conductivity. Why the total K+ + Na+ + Ca2+ is the main contributor? Also it is of strange that DS13 contains less $Fe_2O_3$ than DS12 because DS13 contains biotite 3 times than DS12 and the main Fe carrier in these samples should be biotite. It is better to provide the EPMA data of individual mineral in table 2.

---

## Short Comment (SC1) · 20 Nov 2017

Anonymous Referee 1# This is a much improved submission that most questions have been well answered. I would just want to know how to exclude the effect of iron content on the bulk conductivity. Why the total K+ + Na+ + Ca2+ is the main contributor? Also it is of strange that DS13 contains less Fe2O3 than DS12 because DS13 contains biotite 3 times than DS12 and the main Fe carrier in these samples should be biotite. It is better to provide the EPMA data of individual mineral in table 2. Thanks for your very valuable and

professional comments and suggestions. In the present work, three different gneiss samples were selected to explore the effect of chemical composition on the electrical conductivity under conditions of 623‒1073 K and 0.5‒2.0 GPa. The chemical composition of sample was efficiently controlled by the weight percentage of total content for Na2O + K2O + CaO = 7.12%, 7.27% and 7.64%. According to our obtained results, we found that the electrical conductivities of gneiss samples increased with the rise of the total content of alkali- and calcium ions. As a matter of fact, just as described by the anonymous comments, we designed the initial experimental procedure in order to explore the relationship of hydrous mineral of biotite content influence on the electrical conductivity of gneiss at high temperature and high pressure. However, unfortunately, after we finished a series of conductivity measurements, we did not obtain any available regular change with the content of biotite. All of these obtained results disclosed that the electrical conductivity for gneiss presented a regular variation of the total content of alkali- and calcium ions, which was not related to the content of biotite. According to previously published conductivity results for phlogopite single crystal by Li et al. (2016), they extrapolated that the main charge carriers are probably K+ and F−, and fluorine may play a critical role in electrical conduction. And furthermore, Dai et al. (2014) measured the electrical conductivities of granite with different chemical composition at high temperature and high pressure, and they also adopted the total content of alkali- and calcium ions to establish one functional relationship of electrical conductivity and chemical composition. As we known, the mineralogical assemblages (main rock-bearing minerals are quartz, plagioclase and biotite) between granite and gneiss are almost same. In addition, the activation enthalpies for granite (0.44∼1.18 eV) by Dai et al. (2014) were very approximate to our present obtained results (0.35∼0.87 eV) for the gneiss samples at relevant temperature regions, and the charge carriers of granite were supposed to be K+, Na+ and Ca2+. So, in the present studies, the main contributor for conductivities of gneiss samples is related to K+, Na+ and Ca2+. As for the iron-related small polaron conduction, it is also of one popular conduction type that Fe-bearing silicate minerals and rocks, such as olivine, pyroxene, garnet etc. [e.g. Xu

et al. 2000; Wang et al. 2006; Dai et al. 2009; Yang et al. 2012]. As usual, as a dominant conduction mechanism of small polaron, it is believed that the activation enthalpy is larger than 1.0 eV. In conclusion, it is difficult to extrapolate it as a Fe-related conduction mechanism in the present studies. Indeed, it is possible that the main charge carrier of biotite is the iron-related defect such as the small polaron. In the compilation of this manuscript, according to the optical microscope observation, the biotite content for No DS13 gneiss is close to three times than No DS 12 gneiss, as shown in the mineralogical assemblage of Table 1. However, in light of chemical composition of whole rock analysis by X-ray fluorescence (XRF) in Table 2, the Fe2O3 content of No DS13 gneiss is less than No DS12 gneiss. In consideration of the iron content discrepancy in each biotite, it should be no problem and reasonable. Maybe, if we considered the iron content influence on the electrical conductivity of biotite at high temperature and high pressure, it is one good method of adopting an electronic microprobe analysis to determine the chemical composition. In one previously published paper for gabbro, it is mainly consisted of two dominant mineralogical assemblage (e.g. clinopyroxene and feldspar), and we can also select the X-ray fluorescence (XRF) and electronic microprobe analysis at the same time [Dai et al. 2015]. The XRF and EPMA analysis also were conducted for eclogite in another one our recently published eclogite conductivity with two dominant mineralogical assemblage (e.g. garnet and omphacite) [Dai et al. 2016]. However, in our present work, the mineralogical assemblage is composed of three complex mineralogical assemblage (e.g. quartz, plagioclase and biotite), and it is too complex to acquire any useful information for the explanation of conduction mechanism by EPMA analysis. It is also similar that the influence of chemical composition on the electrical conductivity of granite also only adopted the X-ray fluorescence (XRF) analysis to gain the chemical composition of whole rock [Dai et al. 2014]. And therefore, in the revised manuscript, we did not provide the electronic microprobe analysis results for each individual minerals, and the gneiss sample was considered as a whole to determine the chemical composition influence on its electrical conductivity at high temperature and high pressure.

References Dai, L.D., Hu, H.Y., Li, H.P., Wu, L., Hui, K.S., Jiang, J.J., and Sun, W.Q.: Influence of temperature, pressure, and oxygen fugacity on the electrical conductivity of dry eclogite, and geophysical implications. Geochem. Geophys. Geosyst., 17, 2394–2407, 2016. Dai, L.D., Hu, H.Y., Li, H.P., Hui, K.S., Jiang, J.J., Li, J., and Sun, W.Q.: Electrical conductivity of gabbro: the effects of temperature, pressure and oxygen fugacity. Eur. J. Mineral., 27, 215–224, 2015. Dai, L.D., Hu, H.Y., Li, H.P., Jiang, J.J., and Hui, K.S.: Influence of temperature, pressure, and chemical composition on the electrical conductivity of granite. Am. Mineral., 99, 1420‒1428, 2014. Dai, L.D., and Karato, S.I.: Electrical conductivity of pyrope-rich garnet at high temperature and high pressure. Phys. Earth Planet. Inter., 176, 83‒88, 2009. Li, Y., Yang, X.Z., Yu, J.H., and Cai, Y.F.: Unusually high electrical conductivity of phlogopite: the possible role of fluorine and geophysical implications. Contrib. Mineral. Petrol., 171, 37, 2016. Xu, Y.S, Shankland, T.J., and Duba, A.G.: Pressure effect on electrical conductivity of mantle olivine. Phys. Earth Planet. Inter., 118, 149‒161, 2000. Wang, D.J., Mookherjee, M., Xu, Y.S., and Karato, S.I.: The effect of water on the electrical conductivity of olivine. Nature, 443, 977‒980, 2006. Yang, X.Z., and McCammon, C.: Fe3+-rich augite and high electrical conductivity in the deep lithosphere. Geology, 40, 131‒134, 2012.

Please also note the supplement to this comment:
https://www.solid-earth-discuss.net/se-2017-103/se-2017-103-SC1-supplement.pdf

---

## Referee Comment (RC4) · F. Gaillard (Referee) · 21 Nov 2017

The electrical conductivity of gneiss samples is measured using multi-anvil presses at high-pressure high-temperature. Impedance spectroscopy is used but the paper focuses on the DC results only. The purpose of the paper is to complete a database on the conductivity of crustal rocks with the broad purpose of discussing electrical anomalies in continental crust. Several experimental surveys have been conducted by the same group on different crustal materials, including single crystals. A more specific purpose consists in explaining the Dabie-Sulu ultrahigh-pressure metamorphic belt, in China. This region might be better presented: both the geology and the geophysical

observations deserve a thorough explanation as the reader of Solid Earth is mostly not aware of this area. Regarding the data, we need more information on the run products and on the results: what is the phase proportion? What is (are) the interconnected phase(s) as this is defining the electrical path? Shall we suspect impurities such as carbon or hydrogen to contribute to the DC flow? How these measurements on a multi-phased system compare with the conductivity of individual crystals? How the conductivity compare with other works on, for example, sedimentary gneisses, such as Hashim et al. or Ferri et al? Could the conductivity anomaly in the Dabie-Sulu ultrahigh-pressure metamorphic belt be explained by crustal melting or brines as beneath the Tibetan plateau, on which a vast literature that is ignored here exists? I am looking forward to seeing a ms addressing this issue

---

## Editor Comment (EC1) · U. Werban (Editor) · 21 Nov 2017

postscriptum to RC4 by Fabrice Gaillard added on 21 Nov 2017

The fact that the electrical conductivity increases with increasing pressure most likely indicates that the charge carrier is electronic, not ionic; the authors should investigate this point, which may help them to identify the phase that is conducting in the rock

---

## Author Comment (AC3) · 26 Nov 2017

Response to Professor Fabrice Gaillard: The electrical conductivity of gneiss samples is measured using multi-anvil presses at high-pressure high-temperature. Impedance spectroscopy is used but the paper focuses on the DC results only. The purpose of the paper is to complete a database on the conductivity of crustal rocks with the broad purpose of discussing electrical anomalies in continental crust. Several experimental surveys have been conducted by the same group on different crustal materials, including single crystals. A

more specific purpose consists in explaining the Dabie-Sulu ultrahigh-pressure meta-morphic belt, in China. This region might be better presented: both the geology and the geophysical observations deserve a thorough explanation as the reader of Solid Earth is mostly not aware of this area. Regarding the data, we need more information on the run products and on the results: what is the phase proportion? What is (are) the inter-connected phase(s) as this is defining the electrical path? Shall we suspect impurities such as carbon or hydrogen to contribute to the DC flow? How these measurements on a multi-phased system compare with the conductivity of individual crystals? How the conductivity compare with other works on, for example, sedimentary gneisses, such as Hashim et al. or Ferri et al? Could the conductivity anomaly in the Dabie-Sulu ultrahigh pressure metamorphic belt be explained by crustal melting or brines as beneath the Tibetan plateau, on which a vast literature that is ignored here exists? I am looking forward to seeing a ms addressing this issue. Thanks for your positive comments. I am very appreciated that Professor Fabrice Gaillard for very constructive and enlight-ened comments and suggestions in the reviewing process, which helped us greatly in improving the manuscript. In this revised paper, we conscientiously read through all comments from the valuable suggestions of Professor Fabrice Gaillard, and revised each one points by points, sentences by sentences. All of detailed revisions and re-sponses are listed as follows.

1. Regarding the data, we need more information on the run products and on the results: what is the phase proportion? As shown in table 2, the phase proportion of natural gneiss sample has been provided in detail. The rock-forming minerals of three gneiss samples are feldspar, quartz and biotite, and the contents of the same mineral in each samples are different. Hashim et al. (2013) shows that the dehydration-melting of muscovite starts at 923 K at 0.3 GPa, and biotite is formed in this process. It implies that the mineralogical assemblage of our gneiss samples is stable at a certain range of high temperatures and pressures. Furthermore, it has been confirmed that feldspar, quartz and biotite occur a reaction when T exceeds 1272 K (Ferri et al. 2013). It indicates that the mineralogical assemblage of gneiss is stable at our experimental

temperatures and pressures. Therefore, the phase proportion of the natural sample is same with that of the sample after experiment.

2. What is (are) the interconnected phase(s) as this is defining the electrical path? In the present studies, the rock-forming minerals of our three gneiss samples are feldspar, quartz and biotite, and the volume percentage for each correspondent rock-forming mineral in each gneiss samples were various (Fig.1 and Table 1). The dominant charge carriers of gneiss were proposed to be K+, Na+ and Ca2+. Feldspar is the main mineral with the major elements of K+, Na+ and Ca2+, quartz may contain the impurity ions of K+, Na+ and Ca2+, and biotite contains a certain amount of K+. Therefore, all rock-forming minerals contribute to the conductivities of gneiss. As for the conduction mechanisms for each compositional minerals (feldspar, quartz and biotite) in gneiss, they have been already reported in the previously published work. As pointed by Hu et al. (2011, 2013, 2014, 2015), the main conduction mechanism of feldspar is the alkali- and alkali-Earth ions (e.g. K+, Na+, Ca2+, etc.) by virtue of electrical conductivity measurements and the calculated diffusion coefficient from Nernst–Einstein equation at high temperature and high pressure. The alkali- and alkali-Earth ions, as the dominant charge carriers were transferred between normal lattice alkali positions and adjacent interstitial sites along thermally activated electric fields. Some representative defect reactions for synthetic albite, K-feldspar and anorthite were put forward as follows, (1) (2) (3) The main conduction mechanism in quartz has been investigated in detail by Wang et al., (2010), e.g. the alkali ion moving in channels of crystalline lattice. One typical defect reaction was described as, (4) According to previously published conductivity results for phlogopite single crystal by Li et al. (2016), they extrapolated that the main charge carriers are probably K+ and F−. So, in the present work, we think that some intrinsic defects (e.g. K+, Na+, Ca2+, etc.) in gneiss controlled the main electrical migration path of sample at high temperature and high pressure.

3. Shall we suspect impurities such as carbon or hydrogen to contribute to the DC flow? The conduction mechanism for gneiss sample at high temperature region was

proposed to be intrinsic conduction, but the conduction mechanism at low temperature region was impurity conduction (possible charge carriers: K+, Na+, Ca2+, H+, et al.). It's really possible that carbon or hydrogen contribute to the DC flow.

4. How these measurements on a multi-phased system compare with the conductivity of individual crystals? The mineralogical assemblage of gneiss sample is complicated, and the rock-forming minerals are feldspar, quartz and biotite. Dai et al. (2014) measured the electrical conductivity of granite at 0.5‒1.5 GPa and 623‒1173 K, and the main rock-forming minerals are also quartz, feldspar, and biotite. It was found that the content of calcium and alkali ions significantly influences the electrical conductivities of gneiss. Electrical conductivities of granite and gneiss increase with increasing content of calcium and alkali ions. However, the electrical conductivities of granite were much lower than those of gneiss (Fig. 8). The discrepancy may be caused by the various chemical compositions and rock structure of granite and gneiss. Feldspars are important rock-forming minerals of gneiss, and thus it is important to compare the electrical conductivities of feldspars. The electrical conductivities of alkali feldspars are much higher than the values of the gneiss samples (Hu et al., 2013). It may be due to that the concentrations of alkali ions of alkali feldspars were higher than those of gneisses. In addition, the electrical conductivities of quartz at 1.0 GPa were slightly lower than the values of the gneiss with XA = 7.27% at 1.5 GPa, and the slope of the linear relation between the logarithm of electrical conductivity and the reciprocal of temperature for quartz is close to that for gneiss at lower temperature range (Wang et al. 2010). The conductivities of phlogopite were higher than those of the gneiss with XA = 7.64% at higher temperatures (above 773 K), and lower than those of the gneiss samples at lower temperatures (below 773 K). Furthermore, the slope of the linear relation between the logarithm of electrical conductivity for the phlogopite sample and the reciprocal of temperature is much higher than the slopes for the gneiss samples (Li et al., 2016).

5. How the conductivity compare with other works on, for example, sedimentary

[Figure]

gneisses, such as Hashim et al. or Ferri et al? It's important to compare the conductivities of gneiss with the relevant results of previous studies. As shown in the Fig. 1, the conductivities of the garnet–biotite–sillimanite residual enclave JOY2-X4 are close to the values of gneiss sample DS14 and DS13 at low temperature region and high temperature region, respectively. The conductivities of JOY2-X4 are slightly lower than those of DS12 (Ferri et al. 2013). In addition, the conductivities of natural metapelite PP216 are close to the values of gneiss DS12 at low temperature region, and the slope of relationship between logarithmic conductivities and reciprocal temperature for the metapelite PP216 is higher than those for the gneiss samples at high temperature region (Hashim et al. 2013).

Fig. 1 Comparisons of the electrical conductivities of the gneiss samples measured at 1.5 GPa in this study and in previous studies.

6. Could the conductivity anomaly in the Dabie-Sulu ultrahigh pressure metamorphic belt be explained by crustal melting or brines as beneath the Tibetan plateau, on which a vast literature that is ignored here exists? Thanks for the constructive and enlightened comments and suggestions. Although the conductivities of gneiss samples can't be used to interpret the conductivity anomaly in the Dabie-Sulu ultrahigh pressure metamorphic belt, the conductivity anomaly is probably caused by crustal melting or brines as beneath the Tibetan plateau (Ferri et al. 2013; Hashim et al. 2013). Actually, the geological environment of Dabie-Sulu ultrahigh pressure metamorphic belt is similar to that of the Tibetan plateau. Therefore, the causes for HCLs of two geological units might be similar. Besides, gneiss is widely distributed in the Dabie-Sulu ultrahigh pressure metamorphic belt and Tibetan plateau. Consequently, the conductivity-depth profiles for the gneiss samples with various chemical compositions may provide important constraints on the interpretation of the magnetotelluric results for some regions where the conductivities is close to those of gneiss at high temperatures and pressures.

References Dai, L.D., Hu, H.Y., Li, H.P., Jiang, J.J., and Hui, K.S.: Influence of temperature, pressure, and chemical composition on the electrical conductivity of

granite. Am. Mineral., 99, 1420‒1428, 2014. Hashim, L., Gaillard, F., Champallier, R., Breton, N. L., Arbaret, L., and Scaillet, B.: Experimental assessment of the relationships between electrical resistivity, crustal melting and strain localization beneath the Himalayan-Tibetan Belt. Earth Planet. Sci. Lett., 373, 20‒30, 2013. Hu, H.Y., Li, H.P., Dai, L.D., Shan, S.M., and Zhu, C.M.: Electrical conductivity of albite at high temperatures and high pressures. Am. Mineral., 96, 1821–1827, 2011. Hu, H.Y., Li, H.P., Dai, L.D., Shan, S.M., and Zhu, C.M.: Electrical conductivity of alkali feldspar solid solutions at high temperatures and high pressures. Phys. Chem. Miner., 40, 51‒62, 2013. Hu, H.Y., Dai, L.D., Li, H.P., Jiang, J.J., and Hui, K.S.: Electrical conductivity of K-feldspar at high temperature and high pressure. Mineral. Petrol., 108, 609‒618, 2014. Hu, H.Y., Dai, L.D., Li, H.P., Hui, K.S., and Li, J.: Temperature and pressure dependence of electrical conductivity in synthetic anorthite. Solid State Ionics, 276, 136‒141, 2015. Ferri, F., Gibert, B., Violay, M., and Cesare, B.: Electrical conductivity in a partially molten crust from measurements on metasedimentary enclaves. Tectonophysics, 586, 84‒94, 2013. Li, Y., Yang, X.Z., Yu, J.H., and Cai, Y.F.: Unusually high electrical conductivity of phlogopite: the possible role of fluorine and geophysical implications. Contrib. Mineral. Petrol., 171, 37, 2016. Wang, D.J., Li, H.P., Matsuzaki, T., and Yoshino, T.: Anisotropy of synthetic quartz electrical conductivity at high pressure and temperature. J. Geophys. Res., 115, B09211, doi: 10.1029/2009JB006695, 2010.

Please also note the supplement to this comment:
https://www.solid-earth-discuss.net/se-2017-103/se-2017-103-AC3-supplement.pdf

---

## Author Comment (AC4) · 26 Nov 2017

Response to the editor of Professor Ulrike Werban: The fact that the electrical conductivity increases with increasing pressure most likely indicates that the charge carrier is electronic, not ionic; the authors should investigate this point, which may help them to identify the phase that is conducting in the rock. In the present work, three different gneiss samples were selected to explore the effect of chemical composition on the electrical conductivity under conditions of 623âÅŠ1073 K and 0.5âÅŠ2.0 GPa. The chemical composition of sample

was efficiently controlled by the weight percentage of total content for Na2O + K2O + CaO = 7.12%, 7.27% and 7.64%. According to our obtained results, we found that the electrical conductivities of gneiss samples increased with the rise of the total content of alkali- and calcium ions. Furthermore, we designed the initial experimental procedure in order to explore the relationship of hydrous mineral of biotite content influence on the electrical conductivity of gneiss at high temperature and high pressure. However, unfortunately, after we finished a series of conductivity measurements, we did not obtain any available regular change with the content of biotite. All of these obtained results disclosed that the electrical conductivity for gneiss presented a regular variation of the total content of alkali- and calcium ions, which was not related to the content of biotite. According to previously published conductivity results for phlogopite single crystal by Li et al. (2016), they extrapolated that the main charge carriers are probably  $K_{+}$  and  $F_{-}$ , and fluorine may play a critical role in electrical conduction. And furthermore, Dai et al. (2014) measured the electrical conductivities of granite with different chemical composition at high temperature and high pressure, and they also adopted the total content of alkali- and calcium ions to establish one functional relationship of electrical conductivity and chemical composition. As we known, the mineralogical assemblages (main rock-bearing minerals are quartz, plagioclase and biotite) between granite and gneiss are almost same. In addition, the activation enthalpies for granite  $(0.44 \sim 1.18 \text{ eV})$  by Dai et al. (2014) were very approximate to our present obtained results (0.35~0.87 eV) for the gneiss samples at relevant temperature regions, and the charge carriers of granite were supposed to be K+, Na+ and Ca2+. So, in the present studies, the main contributor for conductivities of gneiss samples is related to K+, Na+ and Ca2+. As for the iron-related small polaron conduction, it is also of one popular conduction type that Fe-bearing silicate minerals and rocks, such as olivine, pyroxene, garnet etc. (Xu et al. 2000; Wang et al. 2006; Dai et al. 2009; Yang et al. 2012). For these Fe-bearing silicate minerals with small polaron conduction, previous studies have confirmed that the conductivities decrease with the increase of pressure (Xu et al. 2000; Dai et al. 2009; Yang et al. 2012). At present studies, the conductivities of the gneiss samples
increased with the rise of pressure. Therefore, we can conclude that the dominant charge carrier for gneiss is possibly not electron but ions.

References Dai, L.D., Hu, H.Y., Li, H.P., Wu, L., Hui, K.S., Jiang, J.J., and Sun, W.Q.: Influence of temperature, pressure, and oxygen fugacity on the electrical conductivity of dry eclogite, and geophysical implications. Geochem. Geophys. Geosyst., 17, 2394-2407, 2016. Dai, L.D., Hu, H.Y., Li, H.P., Hui, K.S., Jiang, J.J., Li, J., and Sun, W.Q.: Electrical conductivity of gabbro: the effects of temperature, pressure and oxygen fugacity. Eur. J. Mineral., 27, 215–224, 2015. Dai, L.D., Hu, H.Y., Li, H.P., Jiang, J.J., and Hui, K.S.: Influence of temperature, pressure, and chemical composition on the electrical conductivity of granite. Am. Mineral., 99, 1420Š1428, 2014. Dai, L.D., and Karato, S.I.: Electrical conductivity of pyrope-rich garnet at high temperature and high pressure. Phys. Earth Planet. Inter., 176, 83âÅŠ88, 2009. Li, Y., Yang, X.Z., Yu, J.H., and Cai, Y.F.: Unusually high electrical conductivity of phlogopite: the possible role of fluorine and geophysical implications. Contrib. Mineral. Petrol., 171, 37, 2016. Xu, Y.S, Shankland, T.J., and Duba, A.G.: Pressure effect on electrical conductivity of mantle olivine. Phys. Earth Planet. Inter., 118, 149aÅŠ161, 2000. Wang, D.J., Mookherjee, M., Xu, Y.S., and Karato, S.I.: The effect of water on the electrical conductivity of olivine. Nature, 443, 977âÅŠ980, 2006. Yang, X.Z., and McCammon, C.: Fe3+-rich augite and high electrical conductivity in the deep lithosphere. Geology, 40, 131âÅŠ134, 2012.

Please also note the supplement to this comment: https://www.solid-earth-discuss.net/se-2017-103/se-2017-103-AC4-supplement.pdf

---

## Author Response (AR1)

**Reponses**

Key Laboratory of High-temperature and High-pressure Study of the Earth's Interior, Institute of Geochemistry, Chinese Academy of Sciences, No 99, Linchengxi Road, Guiyang City, Guizhou Province, P. R. China, 550081 Email address: dailidong@gyig.ac.cn Fax number: 86-0851-85891749 December 28th, 2017

**Dear the editor of Professor Ulrike Werban:**

We have already completed revisions on our manuscript, which is Manuscript Number SE-2017-103 entitled "Effect of chemical composition on the electrical conductivity of gneiss at high temperatures and pressures" by Lidong Dai, Wenqing Sun, Heping Li, Haiying Hu, Lei Wu and Jianjun Jiang, submitted to *Solid Earth*. Above all, we thank the Editor of Professor Ulrike Werban, Professor Fabrice Gaillard and two anonymous reviewers for their very constructive and enlightened comments and suggestions in the reviewing process, which helped us greatly in improving the manuscript. In this revised paper, we conscientiously read through all comments from the editor's and three anonymous reviewers' valuable suggestions, and revised them points by points, sentences by sentences. All of correspondent revisions and responses are listed in the section of Revision Notes.

The editor of Prof. Ulrike Werban and Prof. Fabrice Gaillard claimed that it needs to be greatly clarified for supplementing some substantially profound discussions on the geological motivation and discussion of findings with respect to the recent literature, and as well as identification the charge carrier either electron or ion. Just as commented by the editor in order to make the paper more valuable, we have already rewritten the section of geophysical implication in the revised manuscript. We have already added a large quantity of new detailed descriptions and illustrations on the regional geological background and high conductivity anomaly distribution for the typical ultrahigh-pressure metamorphic (UHPM) belt of Dabie-Sulu region in the discussion section of revised manuscript. A detailed comparison between our obtained gneiss conductivity results with various chemical compositions and field MT data. In consideration of the similar formation conduction and geotectonic environments, the Himalaya–Tibetan orogenic system was compared with the Dabie–Sulu UHPM belt, and explains high electrical conductivity anomalies in detail, and as well as our contribution from the current gneiss with different chemical compositions to be explained more detailedly. Some findings with respect to the most recent literature for the explanation of high conductivity anomaly have already been comprehensively commented and supplemented in the revised manuscript. As for the issue of the charge carrier, we have made one clear elaboration in the Revision Note, and think that the dominant charge carrier for gneiss is possibly not electron but ions.

The 1st anonymous reviewer mainly puts forward three aspects of crucial issues on some detailed sample description (including the possible effect of biotite on the electrical conductivity of gneiss), the possible mechanism of positive pressure effect on the conductivity (including the calculation value for the activation volume) and the supplement for the EPMA data of individual mineral. According to the valuable comments from the first reviewer, we added more detailed sample descriptions in the section of sample preparation, results and discussions in order to declare the issue for the totally alkali- (such as K+ and Na+) and alkali-Earth (Ca2+) metallic ion content on the gneiss conductivity rather than the individual hydrous mineral of biotite. In the revised manuscript, we have already calculated the value of activation volume and presented one clear illustration on the conduction of positive pressure effect. As for the issue of supplement for the EPMA data of individual mineral, some detailed explanations have been supplemented in the Revision Notes.

The 2nd anonymous reviewer mainly concerns the issue on the real and imaginary parts of ac-conductivity and impedance as a function of the measured frequency at different T and P, except of the Cole-Cole plots of complex impedance, and as well as the fitted equivalent circuit of impedance spectroscopy. In the revised paper, we supplemented one absolutely new Figure 4 so as to display the real and imaginary parts of complex impedance for runs 13 and 14 gneiss samples as a function of the measured frequency at 1.5 GPa and 623-1073 K. At the same time, some further detailed explanations have already been added in the revised manuscript. As for the issue of the fitted equivalent circuit of impedance spectroscopy, the bulk sample resistance can be determined by fitting the high-frequency semicircular arc. The fitted equivalent circuit is composed of the series connection of  $R_S$ –CPES ( $R_S$  and CPES represent the resistance and constant-phase element of a sample, respectively) and  $R_E$ –CPEE ( $R_E$  and CPEE represent the interaction of the charge carrier with the electrode).we have already changed the capacitor ( $C_s$ ) into the constant phase element (CPE) in the revised manuscript.

The 3rd reviewer of Prof. Fabrice Gaillard mainly concerns the issue on geological and geophysical observations deserving a thorough explanation for the Dabie-Sulu ultrahigh-pressure metamorphic belt, electrical path, a comparison with other previous works for the sedimentary gneisses, and the conductivity anomaly in the Dabie-Sulu ultrahigh pressure metamorphic belt explained by crustal melting or brines as beneath the Tibetan plateau. In the revised manuscript, a large amount of thoroughly and substantially geological and geophysical observations for the Dabie-Sulu ultrahigh-pressure metamorphic belt have already been supplemented. In the present work, we think that some intrinsic defects (e.g.  $K^+$ ,  $Na^+$ ,  $Ca^{2+}$ , etc.) in gneiss controlled the main electrical migration path of sample at high temperature and high pressure. A detailed comparison with other previous works for the sedimentary gneisses has already been supplemented in the revised Figure 8 and context of the revised manuscript. In consideration of the similar formation and geological tectonic environments, the Himalaya-Tibetan orogenic system was selected to be compared and explained the cause of high electrical conductivity anormaly for the Dabie-Sulu UHPM belt in detail, and the high conductivity anomaly in the Dabie-Sulu ultrahigh pressure metamorphic belt was explained by the interconnected fluids or melts.

In addition, according to the precious comments and suggestions from the Editor of Professor Ulrike Werban, Professor Fabrice Gaillard and two anonymous reviewers, we have already added two absolutely new Figures 4 and 9, and replotted Figures 3 and 8, and rechecked each listed diagrams and tables in the revised paper very conscientiously. All of corresponding contexts and references have already been rechecked and modified thoroughly in the revised paper. As for the paper's English writing style and expression skills, we appreciate Dr Kara Bogus from Edanz Group (www.edanzediting.com/ac) Scientific Editing Company for their helps in English improvements of the manuscript. The substantial corrections for English have been conducted sentences by sentences. After that, the revised paper becomes much more easily be read and understood.

Thank you very much again for many kind comments and advisements from the Editor of Professor Ulrike Werban, Professor Fabrice Gaillard and two anonymous reviewers to put forward a large amount of crucial and constructive suggestions to greatly improve our manuscript. In sum, we made great efforts answering all of these questions one by one opinions, and revising the manuscript points by points, sentences by sentences, accordingly. As for the paper's English writing style and expression skills, we appreciate Dr Kara Bogus from Edanz Group (www.edanzediting.com/ac) Scientific Editing Company for their helps in English improvements of the manuscript. We believe that the revised manuscript has been significantly improved, and hope it is now acceptable for your publication in *Solid Earth*.

With best Regards,

Lidong Dai, PhD, Corresponding author

**Revision Notes**

**Response to the editor of Professor Ulrike Werban:**

1. The fact that the electrical conductivity increases with increasing pressure most likely indicates that the charge carrier is electronic, not ionic; the authors should investigate this point, which may help them to identify the phase that is conducting in the rock.

In the present work, three different gneiss samples were selected to explore the effect of chemical composition on the electrical conductivity under conditions of 623–1073 K and 0.5–2.0 GPa. The chemical composition of sample was efficiently controlled by the weight percentage of total content for  $Na_2O + K_2O + CaO = 7.12\%$ , 7.27% and 7.64%. According to our obtained results, we found that the electrical conductivities of gneiss samples increased with the rise of the total content of alkaliand calcium ions. Furthermore, we designed the initial experimental procedure in order to explore the relationship of hydrous mineral of biotite content influence on the electrical conductivity of gneiss at high temperature and high pressure. However, unfortunately, after we finished a series of conductivity measurements, we did not obtain any available regular change with the content of biotite. All of these obtained results disclosed that the electrical conductivity for gneiss presented a regular variation of the total content of alkali- and calcium ions, which was not related to the content of biotite. According to previously published conductivity results for phlogopite single crystal by Li et al. (2016), they extrapolated that the main charge carriers are probably K+ and F-, and fluorine may play a critical role in electrical conduction. And furthermore, Dai et al. (2014) measured the electrical conductivities of granite with different chemical composition at high temperature and high pressure, and they also adopted the total content of alkali- and calcium ions to establish one functional relationship of electrical conductivity and chemical composition. As we known, the mineralogical assemblages (main rock-bearing minerals are quartz, plagioclase and biotite) between granite and gneiss are almost same. In addition, the

activation enthalpies for granite (0.44~1.18 eV) by Dai et al. (2014) were very approximate to our present obtained results (0.35~0.87 eV) for the gneiss samples at relevant temperature regions, and the charge carriers of granite were supposed to be K+, Na+ and Ca2+. So, in the present studies, the main contributor for conductivities of gneiss samples is related to K+, Na+ and Ca2+. As for the iron-related small polaron conduction, it is also of one popular conduction type that Fe-bearing silicate minerals and rocks, such as olivine, pyroxene, garnet etc. (Xu et al., 2000; Wang et al., 2006; Dai et al., 2009; Yang et al., 2012). For these Fe-bearing silicate minerals with small polaron conduction, previous studies have confirmed that the conductivities decrease with the increase of pressure (Xu et al., 2000; Dai et al., 2009; Yang and McCammon, 2012). At present studies, the conductivities of the gneiss samples increased with the rise of pressure. Therefore, we can conclude that the dominant charge carrier for gneiss is possibly not electron but ions.

2. Many thanks for submission of your article (SE-2017-103, Effect of chemical composition on the electrical conductivity of gneiss at high temperatures and pressures) and providing comments to the issues raised by the reviewer. I went through the referee's comments and have to admit that the comments went clear in the direction of a rejection, since the manuscript needs very substantial revision. I can follow your statements and some of the issues raised by the reviewers where addressed whereas others still are open. Especially the comments from reviewer 3 concerning the geophysical and geological implication (provide more details on the geological background and previous discussions and your contribution to explain the Dabie-Sulu ultrahigh-pressure meta-morphic belt) are still open. Please keep in mind that SE is an inter- and multidisciplinary journal and the reader is not only interested in the pure experiment but also in the motivation and discussion of findings with respect to the recent literature. Just presenting a new dataset is not enough to fullfil the requirements of the journal. Going through you comments I also realized that many issues that could be discussed in this paper and make the paper valuable are already addressed in other recent publications of Dai et al.. This fact enhances the need for a clear distinction from the other papers and as already mentioned the presentation of a new dataset is not enough from my point of view. I recommend to submit a revised version only if you are sure that you address all the issues raised in the discussions which means substantial changes in the manuscript and additional profound discussion on the geological motivation. For only presenting the dataset another journal might be more appropriate in this case.

Thanks for your very constructive and enlightened comments and suggestions in the reviewing process, which helped us greatly in improving the manuscript. In this revised paper, we conscientiously read through all comments from the valuable suggestions of the editor and three anonymous reviewers, and revised them one points by points, sentences by sentences. A large amount of thorough and substantial revisions have been already conducted for our manuscript. Just as commented by the editor of Professor Ulrike Werban, it is indeed crucial to deeply discuss the geophysical and geological implications for electrical conductivities of gneiss at high temperature and high pressure. And thus, in order to clearly declare the issue of geophysical and geological implications, we have already supplemented a large quantity of detailed descriptions and illustrations on the regional geological background for the typical ultrahigh-pressure metamorphic (UHPM) belt of Dabie-Sulu region in the discussion section of revised manuscript. In the present studies, although the conductivity results on gneiss can't be used to interpret the high conductivity anomaly in the Dabie-Sulu UHPM belt, the conductivity-depth profiles we have constructed for gneiss with different chemical compositions may provide important constraints on the interpretation for field magnetotelluric conductivity in the regional UHPM belt. In consideration of the similar formation conduction and geotectonic environments, the Himalaya-Tibetan orogenic system was compared with the Dabie-Sulu UHPM belt, and explains high electrical conductivity anomalies in detail, and as well as our contribution from the current gneiss with different chemical compositions to be explained more detailedly.

[revised manuscript text omitted]
 biotite-bearing felsic gneiss at high P-T conditions. They tried to explain the conductivity differences by the contribution of total  $K^++Na^++Ca^{2+}$  of three natural gneiss samples. The experimental technique is top-notch but the strategy and discussion are not convincing.

Thanks for the positive comments. In this revised manuscript, we conscientiously read through all valuable comments and suggestions, and revised each one points by points, sentences by sentences. So far we have made some substantial strategy and discussion convinced in the revised manuscript.

1. I think the manuscript must be revised largely and more evidences should be provided before publication. The authors measured the electrical conductivity of gneiss parallel to foliation. There are at least two reasons may contribute to the conductivity differences, including chemical composition effect and textural difference. How to evaluate the effect of textures? Biotite usually deforms and aggregates to form the band texture and it may exhibit strong conductivity anisotropy, highest along the layered surface and lowest perpendicular to the layered structure. The conductivity differences, therefore, may result from the texture differences. The authors did not describe the samples carefully.

Thanks for your valuable and professional comments and suggestions. Indeed, just as described by the first anonymous reviewer, it is possibly existing two dominant reasons of chemical composition and texture that can result in the difference of electrical conductivity measurement results. Based on the results of the previously reported studies, the main conduction mechanism for phlogopite is ionic conduction, and K+ is proposed to be the main charge carriers (Li et al., 2017a, b). We suggested that the charge carriers of the gneiss samples were K+, Na+ and Ca2+. Therefore, the influence of biotite on the conductivities of gneiss has been taken into consideration. On the other hand, the electrical conductivities of the gneiss samples don't regularly

increase with increasing content of biotite, as shown in Table 1 and Fig. 6. Based on all of these obtained experimental results, it made clear that the content of biotite is not the main influence factor influence on the electrical conductivity of gneiss samples. In the present studies, we considered the gneiss sample as a whole to explore its electrical conductivity at high temperature and high pressure, and it is crucial that the chemical composition of sample ( $W_A=Na_2O+K_2O+CaO=7.12\%$ , 7.27% and 7.64% in weight percent, respectively) is really a significant influence on the electrical conductivity of sample. We find that the electrical conductivities of gneiss samples dramatically increase with the rise of  $W_A$ .

On the base of the valuable suggestion from the anonymous reviewer, we have already supplemented a large quantity of detailed description in the section of 2.1 sample preparation in the revised manuscript. Some main revisions have been summarized as follows:

Three relatively homogeneous natural gneiss samples with a parallel to foliation direction were collected from Xinjiang, China. The surface of the sample is fresh, non-fractured and non-oxidized, without evidence of alteration before and after experiments. The main rock-forming minerals of three gneiss samples are feldspar, quartz and biotite, respectively. It was indicated that three gneiss samples have the same mineralogical assemblage, and all of them belong to biotite-bearing felsic gneiss. From Table 2, we found that the totally alkali- (such as K+ and Na+) and alkali-Earth (Ca2+) metallic ion content for each sample were various. And therefore, in the present studies, we have conducted a series of experiments in order to determine the influence of chemical composition by changing the totally alkali- and alkali-Earth metallic ion content on the electrical conductivity of gneiss at high temperature and high pressure.

2. Even that the effect of chemical compositions dominates on the conductivities, the authors cannot use the composition data of a whole rock as that of the unique sample used in conductivity measurement because of the inhomogeneity. To overcome these uncertainties, well mixed powder samples must be used instead although the geological application will be penalized.

Thanks for your professional comments and advisements. Indeed, it is one inevitable problem of the sample's inhomogeneity only if the researcher tried to measure the electrical conductivity of natural rock at high temperature and high pressure. Just as described by the anonymous reviewer, it's true that chemical composition for hot-pressed sintering sample by the mixed powder samples seems much more homogeneous than those of natural samples. In this study, we chose a series of natural samples rather than hot-pressed sintering sample, mainly considered: (a) the structure of mixed powder sample is completely different from that of natural sample, which implies that the natural sample become more representative to explore its geophysical implications; (b) In the process of hot-pressed sintering sample, grain size is difficult to control for each experiment, and therefore, the grain size influence on the electrical conductivity issue for one complex rock is not easy to be well solved; (c) Only if one natural rock sample of its mineralogical assembly contained one or several hydrous minerals, such as amphibole, mica et al., it is not strongly suggested that we chose one hot-pressed sintering method to synthesize the starting experimental sample. If the hot-pressed temperature is too low, I am afraid that some inevitable fractures and microcrackings have some influences on the subsequent electrical conductivity measurement. On the contrary, if the hot-pressed temperature is too high, the dehydration of hydrous mineral must be full considered in the process of sample preparation. As a matter of fact, in our previously reported papers, we have already completed electrical conductivity measurements on many representative natural rock samples at high temperature and high pressure in our laboratory, such as natural samples: pyroxenite (Dai et al., 2006), lherzolite (Dai et al., 2008), amphibolite (Zhou et al., 2011; Wang et al., 2012), granite (Dai et al., 2014), basalt (Dai et al., 2015), gabbro (Dai et al., 2015), and eclogite (Dai et al., 2016), etc. In addition, much more papers on the electrical conductivity of natural rocks have been also published in other laboratory, such as granulite (Fuji-ta et al., 2004), gneiss (Fuji-ta et al., 2007), amphibolite (Saltas et al., 2013), and quartzite (Shimojuku et al., 2014), etc.

In addition, we made great efforts in choosing small area of three relatively

homogeneous natural gneiss samples with a parallel to foliation direction in the process of our current sample preparation. During the conductivity measurements, we cut and polish them into a cylinder of  $\Phi$  6.0 × 6.0 mm in order to efficiently avoid this issue. Of course, in the future, we can try to measure one hot-pressed synthetic gneiss sample and compare it.

3. It is also a strange strategy that the authors haven't choose the samples from Dabie-Sulu as the starting materials, despite finally they apply the results to explain the HCL within Dabie-Sulu.

Thanks for your valuable comments. To be frank, due to some practical difficulties for our own work area, we didn't collect a series of natural gneiss samples originated from the region of Dabie-Sulu ultrahigh-pressure metamorphic belt. However, it has been confirmed that abundant felsic gneisses were widespread distributed in Dabie-Sulu ultrahigh-pressure metamorphic belt, and the mineralogical assemblage of gneiss in Dabie-Sulu ultrahigh-pressure metamorphic belt is similar to that of our present experimental samples (Gong et al., 2013). In addition, the gneiss distributed in the deep Earth interior may be existing some discrepancy from that of outcrop in the Earth's surface. Three gneisses with various chemical compositions are able to represent many natural biotite-bearing felsic gneiss, and we arrived in one conclusion that the electrical conductivities of gneiss cannot be used to interpret the high conductivity layers (HCLs) in Dabie-Sulu ultrahigh-pressure metamorphic belt.

**Other comments:**

(1) Quality of writing: In its present state, this article is not publishable. Writing needs tremendous improvements to match the requirements of any peer-reviewed journal.

As for the paper's English writing style and expression skills, we appreciate Dr Kara Bogus from Edanz Group (www.edanzediting.com/ac) Scientific Editing Company for their helps in English improvements of the manuscript. The substantial corrections for English have been conducted sentences by sentences. After that, the revised paper becomes much more easily be read and understood.

(2) The authors should calculate the activation volume for Run DS12, and explain the possible mechanism of positive pressure effect on the conductivity.

According to the suggestion, we have already supplemented all of these results on the activation volume for Run DS12 and the calculating equation. With increasing pressure, the electrical conductivity of gneiss increases, accordingly. The activation volumes for Run DS12 are -7.10 cm3/mole and -2.69 cm3/mole at low temperature region and high temperature region, respectively. Another one representative metamorphic rock for gneiss, we can compared it with the electrical conductivity of eclogite. Recently, Dai et al. (2016) measured the electrical conductivity of dry eclogite, and the obtained negative activation volume value for eclogite is -2.51 cm3/mole under conditions of 1.0-3.0 GPa and 873-1173 K. It was proposed that the main conduction mechanism for dry eclogite is intrinsic conduction (Dai et al., 2016). The conduction mechanism for gneiss sample at high temperature region was also proposed to be intrinsic conduction, but the conduction mechanism at low temperature region was impurity conduction (possible charge carriers: K+, Na+, Ca2+, H+, et al.). In addition, it was suggested that the positive pressure effect on the electrical conductivities of gneiss samples may be due to the more complicated rock structure.

(3) Line 322-325: The authors should clearly show how to convert the conductivity temperature data to conductivity-depth profile with the aid of heat flow for the general readers.

Thanks for your professional and precious suggestions. The relationship between temperature and depth in the Earth's stationary crust can be obtained by a numerical solution of the heat conduction equation (Selway et al., 2014):

$$T = T_0 + (\frac{Q}{k})Z - (\frac{A_0}{2k})Z^2$$
(3)

where  $T_0$  is the surface temperature (K), Q is the surface heat flow (mW/m2), Z is the

lithosphere layer depth (km), k is thermal conductivity (W/mK), and  $A_0$  is the lithospheric radiogenic heat productivity ( $\mu$ W/m3). Based on previous studies, the corresponding thermal calculation parameters for the Dabie–Sulu orogen are Q=5 mW/m2 (He et al., 2009),  $A_0=0.31 \mu$ W/m3 and k=2.6 W/mK (Zhou et al., 2011).

Based on the heat conduction equation (Eq. 3) and thermal calculation parameters, the conductivity-temperature results of gneiss with various chemical compositions ( $W_A=Na_2O+K_2O+CaO=7.12\%$ , 7.27% and 7.64%) can be converted to a conductivity-depth profile for the Dabie–Sulu orogen.

2. This is a much improved submission that most questions have been well answered. I would just want to know how to exclude the effect of iron content on the bulk conductivity. Why the total K+ + Na+ + Ca2+ is the main contributor? Also it is of strange that DS13 contains less Fe2O3 than DS12 because DS13 contains biotite 3 times than DS12 and the main Fe carrier in these samples should be biotite. It is better to provide the EPMA data of individual mineral in table 2.

Thanks for your very valuable and professional comments and suggestions. In the present work, three different gneiss samples were selected to explore the effect of chemical composition on the electrical conductivity under conditions of 623-1073 K and 0.5-2.0 GPa. The chemical composition of sample was efficiently controlled by the weight percentage of total content for Na2O+K2O+CaO=7.12%, 7.27% and 7.64%. According to our obtained results, we found that the electrical conductivities of gneiss samples increased with the rise of the total content of alkali- and calcium ions. As a matter of fact, just as described by the anonymous comments, we designed the initial experimental procedure in order to explore the relationship of hydrous mineral of biotite content influence on the electrical conductivity of gneiss at high temperature and high pressure. However, unfortunately, after we finished a series of conductivity measurements, we did not obtain any available regular change with the content of biotite. All of these obtained results disclosed that the electrical conductivity for gneiss presented a regular variation of the total content of alkali- and calcium ions, which was not related to the content of biotite. According to previously published

conductivity results for phlogopite single crystal by Li et al. (2016), they extrapolated that the main charge carriers are probably K+ and F-, and fluorine may play a critical role in electrical conduction. And furthermore, Dai et al. (2014) measured the electrical conductivities of granite with different chemical composition at high temperature and high pressure, and they also adopted the total content of alkali- and calcium ions to establish one functional relationship of electrical conductivity and chemical composition. As we known, the mineralogical assemblages (main rock-bearing minerals are quartz, plagioclase and biotite) between granite and gneiss are almost same. In addition, the activation enthalpies for granite (0.44~1.18 eV) by Dai et al. (2014) were very approximate to our present obtained results (0.35~0.87 eV) for the gneiss samples at relevant temperature regions, and the charge carriers of granite were supposed to be  $K^+$ ,  $Na^+$  and  $Ca^{2+}$ . So, in the present studies, the main contributor for conductivities of gneiss samples is related to K+, Na+ and Ca2+. As for the iron-related small polaron conduction, it is also of one popular conduction type that Fe-bearing silicate minerals and rocks, such as olivine, pyroxene, garnet etc. (e.g. Xu et al., 2000; Wang et al., 2006; Dai et al., 2009; Yang et al., 2012). As usual, as a dominant conduction mechanism of small polaron, it is believed that the activation enthalpy is larger than 1.0 eV. In conclusion, it is difficult to extrapolate it as a Fe-related conduction mechanism in the present studies.

Indeed, it is possible that the main charge carrier of biotite is the iron-related defect such as the small polaron. In the compilation of this manuscript, according to the optical microscope observation, the biotite content for No DS13 gneiss is close to three times than No DS 12 gneiss, as shown in the mineralogical assemblage of Table 1. However, in light of chemical composition of whole rock analysis by X-ray fluorescence (XRF) in Table 2, the Fe2O3 content of No DS13 gneiss is less than No DS12 gneiss. In consideration of the iron content discrepancy in each biotite, it should be no problem and reasonable. Maybe, if we considered the iron content influence on the electrical conductivity of biotite at high temperature and high pressure, it is one good method of adopting an electronic microprobe analysis to determine the chemical composition. In one previously published paper for gabbro, it is mainly consisted of

two dominant mineralogical assemblage (e.g. clinopyroxene and feldspar), and we can also select the X-ray fluorescence (XRF) and electronic microprobe analysis at the same time (Dai et al., 2015). The XRF and EPMA analysis also were conducted for eclogite in another one our recently published eclogite conductivity with two dominant mineralogical assemblage (e.g. garnet and omphacite) (Dai et al., 2016). However, in our present work, the mineralogical assemblage is composed of three complex mineralogical assemblage (e.g. quartz, plagioclase and biotite), and it is too complex to acquire any useful information for the explanation of conduction mechanism by EPMA analysis. It is also similar that the influence of chemical composition on the electrical conductivity of granite also only adopted the X-ray fluorescence (XRF) analysis to gain the chemical composition of whole rock (Dai et al., 2014). And therefore, in the revised manuscript, we did not provide the electronic microprobe analysis results for each individual minerals, and the gneiss sample was considered as a whole to determine the chemical composition influence on its electrical conductivity at high temperature and high pressure.

**Response to the anonymous Reviewer 2#:**

In their submitted manuscript the authors investigate the electrical properties of different gneiss samples at elevated temperatures and high hydrostatic pressures by means of state of the art experimental facilities. The paper focuses on the effect of the chemical composition to the measured conductivity and different conduction mechanisms are reported. Geophysical implication is also discussed. The work is interesting and worth publishing but additional aspects could also be revealed after further analysis of the experimental data. The authors should pay much effort to improve the quality of their work, in order to be suitable for publication. The following issues should be carefully addressed:

We thank the anonymous reviewer for very constructive and enlightened comments and suggestions in the reviewing process, which helped us greatly in improving the manuscript. In this revised paper, we conscientiously read through all comments from the valuable suggestions of the reviewer, and revised each one points by points, sentences by sentences. All of detailed revisions and responses are listed as follows.

1. In my opinion, the author should not just limited to the calculations of the dc-conductivity but also explore the advantages of the complex impedance spectroscopy. Otherwise, they could measure the dc-conductivity by varying linearly the temperature at different selected pressures. I suggest using also other formalisms of impedance data, such as ac-conductivity and complex impedance presentation of their data.

Thanks for your valuable and professional comments and suggestions. As a matter of fact, it is indeed one good idea that we can calculate the complex impedance presentation and ac-conductivity from the complex impedance spectroscopy. In the revised manuscript, we have already supplemented another one Figure 4: Real and imaginary parts of complex impedance as functions of the measured frequencies for run DS13 and DS14 gneiss under conditions of 1.5 GPa and 623–1073 K. (a) real and (b) imaginary parts for run DS13 gneiss; (c) real and (d) imaginary parts for run DS14 gneiss.

In order to explore the geophysical implication from the electrical conductivities of gneiss samples, we calculated the dc-conductivities of natural gneiss samples, and researched the influence of chemical compositions, temperatures and pressures. Indeed, most previous studies calculated dc-conductivities of minerals and rocks to compare with Magnetotelluric (MT) and geomagnetic depth sounding (GDS) results (Fuji-ta et al., 2007; Dai et al., 2016; Hu et al., 2017).

2. According to my previous comment, it would be also desirable to present the results of all the measured samples (or at least of 2 of them) in suitable figures, i.e. real and imaginary parts of ac-conductivity and impedance as a function of the measured frequency at different T and P, except of the Cole-Cole plots of complex impedance.

Thanks for your valuable and professional comments and suggestions. According the anonymous reviewer's suggestion, we added one absolutely new diagram about the real and imaginary parts of complex impedance for the runs DS13 and DS14 gneiss samples as a function of the measured frequency at 1.5 GPa and 623–1073 K. From the real and imaginary parts of complex impedance as functions of the measured frequencies (Figure 4), the real part values almost remain unchanged over a frequency range of  $10^{6}$ – $10^{4}$  Hz, and sharply increased at  $10^{4}$ – $10^{2}$  Hz; these values then slowly increased within the  $10^{2}$  to  $10^{-1}$  Hz lower frequency range of  $10^{6}$ – $10^{5}$  Hz, the values gradually increased at  $10^{5}$ – $10^{3}$  Hz and decreased at  $10^{3}$ – $10^{1}$  Hz; and these values then slowly increased in the  $10^{1}$  to  $10^{-1}$  Hz lower frequency region.

---

## Author Response (AR2)

**Responses**

Key Laboratory of High-temperature and High-pressure Study of the Earth's Interior, Institute of Geochemistry, Chinese Academy of Sciences, No 99, Linchengxi Road, Guiyang City, Guizhou Province, P. R. China, 550081 Email address: dailidong@gyig.ac.cn Fax number: 86-0851-85891749 February 6th, 2018

**Dear the editor of Professor Ulrike Werban:**

We have already completed revisions on our manuscript, which is Manuscript Number SE-2017-103 entitled "Effect of chemical composition on the electrical conductivity of gneiss at high temperatures and pressures" by Lidong Dai, Wenqing Sun, Heping Li, Haiying Hu, Lei Wu and Jianjun Jiang, submitted to *Solid Earth*. Above all, we thank the Editor of Professor Ulrike Werban, Professor Fabrice Gaillard and two anonymous reviewers for their very constructive and enlightened comments and advisements in the reviewing process, which helped us greatly in improving the manuscript. In this revised paper, we conscientiously read through all comments from the editor's and three anonymous reviewers' valuable suggestions, and revised them points by points, sentences by sentences. All of correspondent revisions and responses are listed in the section of Revision Notes.

Thank you very much again for many kind comments and suggestions from the Editor of Professor Ulrike Werban, Professor Fabrice Gaillard and two anonymous reviewers to put forward a large amount of crucial and constructive suggestions to greatly improve our manuscript. We made great efforts answering all of these questions one by one opinions, and revising the manuscript points by points, sentences by sentences, accordingly. The revised manuscript has been significantly improved, and hope it is now acceptable for your publication in *Solid Earth*.

With best Regards,

Lidong Dai, PhD, Corresponding author

**Revision Notes**

**Response to the anonymous Reviewer 1#:**

The manuscript has been consolidated significantly. I now just have some less important comments.

1. The descriptions in lines 74-75 are not pertinent. Why the protolith of gneiss is granite? This is basically incorrect. The description should be careful.

Thanks for your valuable comments. Indeed, just as described by the first anonymous reviewer, as a complex rock for gneiss, it is not related with our present title "Effect of chemical composition on the electrical conductivity of gneiss at high temperatures and pressures". And therefore, we removed the content of descriptions in lines 74–75 in the revised manuscript.

2. I think "Therefore, in the... by changing the total alkali and calcium ion content" is not necessary and also not suitable. There are total 3 experiments. All the samples are just natural ones, no special efforts are adopted to synthesize samples with different alkali composition. The experimental results can be, but maybe occasionally, explained by the total alkali and calcium components. Therefore, not necessary to emphasize that you measure the conductivity by changing the total alkali and calcium ion content.

Thanks for your valuable and professional comments and suggestions. In the present studies, the electrical conductivity was conducted for three natural gneisses with different alkali composition rather than hot-pressed synthetic samples in main consideration that the natural rock samples become more representative to explore its geophysical implications. Of course, in the section of sample preparation, the sentence is a little duplicated and was deleted in the revised manuscript.

**3.* What is $F^+$ in line 287?**

Thanks for your conscientious corrections. I have already corrected F+ into F- in

line 287.

**4. The activation energy from 0.5 to 2.0 GPa should be stated in Table 3.**

Thanks for your valuable comments. According to your precious comments, we have already supplemented the activation energy values at the pressure range of 0.5 to 2.0 GPa in Table 3 and the calculated equation in the context of the revised manuscript.

5. What kind of difference in mineralogical assemblage and chemical composition between gneiss and granite in line 326-327? The authors should show it rather than just mention it.

Thanks for your valuable comments. The main minerals of our gneiss samples are plagioclase, quartz and biotite (Table 1). Dai et al. (2014) studied the electrical conductivity of natural granite with various chemical compositions at 623-1173 K and 0.5 GPa. The main rock-forming minerals of the granite sample are also plagioclase, quartz and biotite. However, the biotite content of the granite sample is smaller than the content of gneiss. For chemical compositions, the contents of SiO2 in the gneiss samples are lower than those in the granite samples; the contents of calc-alkali ions in the gneiss samples are close to those in the granite samples. Therefore, the dependence of electrical conductivity of gneiss on chemical composition is not identical to granite.

6. Are there studies to interpret the high conductivity anomalies within Dabie-Sulu by fluids or melts?

Thanks for your valuable and professional comments and suggestions. From the field geophysical observations, previously magnetotelluric results have already confirmed that it is widely existed the high conductivity anomalies within Dabie–Sulu ultra-high pressure metamorphic belt and interpreted the cause of the high conductivity anomalies by partial melting and water–bearing (or saline–bearing) fluids (Xiao et al., 2007). From the viewpoint of geochemistry, some stable isotope

geochemical evidences have also disclosed that the water-bearing (or saline-bearing) fluids and partial melting phenomena were widely observed in the Dabie-Sulu ultra-high pressure metamorphic belt (e.g. Zheng et al., 2003; Zhao, Z., and Zheng, Y.: Remelting of subducted continental lithosphere: Petrogenesis of Mesozoic magmatic rocks in the Dabie-Sulu orogenic belt. Science in China Series D: Earth Sciences, 52, 1295–1318, 2009). In comprehensive consideration of Professor Fabrice Gaillard's comments, the Himalaya–Tibetan orogenic system with similar formation conduction and geotectonic environments was selected to compare it with the Dabie–Sulu UHPM belt in order to explain the high electrical conductivity anomalies for Dabie–Sulu ultrahigh pressure metamorphic belt. However, the direct experimental evidences of the laboratory-based high-pressure measurements for the interpretation of the high conductivity anomalies within Dabie–Sulu ultrahigh pressure metamorphic belt by fluids or melts are scarce.

**Response to Professor Fabrice Gaillard:**

1. I think the author have conducted a great deal of effort to change the ms. Only minor typos or mistake are spread within the ms: (eg. line 45: of instead of "for"; line 49: number instead of "quantity"; remove the 's line 63 of use mineralogical assemblage of granulite...).

Thanks for your conscientious corrections. I have already corrected them one by one very carefully in the revised manuscript.

2. I don't like the new title as "complex impedance spectroscopy" is not speaking to anyone.

Thanks for Professor Fabrice Gaillard's valuable and professional comments. We have changed the title into the initial one "Effect of chemical composition on the electrical conductivity of gneiss at high temperatures and pressures".

3. Regarding the last sentence that the high conductivity at Dabie-Sulu may be due to deep fluids or melts, is there any geological evidence for a magmatic activity in this

**area (eg granitic rocks or volcanoes)?**

Thanks for your valuable and professional comments and suggestions. From the field geophysical observations, previously magnetotelluric results have already confirmed that it is widely existed the high conductivity anomalies within Dabie-Sulu ultra-high pressure metamorphic belt and interpreted the cause of the high conductivity anomalies by partial melting and water-bearing (or saline-bearing) fluids (Xiao et al., 2007). From the viewpoint of geochemistry, some stable isotope geochemical evidences have also disclosed that the water-bearing (or saline-bearing) fluids and partial melting phenomena were widely observed in the Dabie-Sulu ultra-high pressure metamorphic belt (e.g. Zheng et al., 2003; Zhao, Z., and Zheng, Y.: Remelting of subducted continental lithosphere: Petrogenesis of Mesozoic magmatic rocks in the Dabie-Sulu orogenic belt. Science in China Series D: Earth Sciences, 52, 1295-1318, 2009). In comprehensive consideration of Professor Fabrice Gaillard's comments, the Himalaya-Tibetan orogenic system with similar formation conduction and geotectonic environments was selected to compare it with the Dabie-Sulu UHPM belt in order to explain the high electrical conductivity anomalies for Dabie–Sulu ultrahigh pressure metamorphic belt. However, the direct experimental evidences of the laboratory-based high-pressure measurements for the interpretation of the high conductivity anomalies within Dabie-Sulu ultrahigh pressure metamorphic belt by fluids or melts are scarce.

**4. Figure 8: please try to make the symbols and lines for this study easier to see.**

Thanks for your precious comments and suggestions. We have already tried my best to reedit, decorate and adjust Figure 8 that the symbols and as well as the artistic contrast of line in the shapes and color make it more clear and easily understood in the revised manuscript.

5. In fig 10 at 30 km the gneisses reach the electrical conductivity of 0.1 S.m, which corresponds to an electrical anomaly; what is the temperature at such a depth, then? is it possible to have such a temperature at 15 km? Would it implies melting?

Thanks for your valuable and professional comments and suggestions. According to previous study, the relationship between temperature and depth in the Earth's stationary crust can be obtained by a numerical solution of the heat conduction equation (Selway et al., 2014):

$$T = T_0 + (\frac{Q}{k})Z - (\frac{A_0}{2k})Z^2$$
(4)

where  $T_0$  is the surface temperature (K), Q is the surface heat flow (mW/m2), Z is the lithosphere layer depth (km), k is thermal conductivity (W/mK), and  $A_0$  is the lithospheric radiogenic heat productivity ( $\mu W/m^3$ ). Based on previous studies, the corresponding thermal calculation parameters for the Dabie–Sulu orogen are Q=75mW/m2 (He et al., 2009),  $A_0=0.31 \mu$ W/m3 and k=2.6 W/mK (Zhou et al., 2011). Based on the heat conduction equation and the corresponding thermal calculation parameters for the Dabie–Sulu orogen, the depths of 30 km and 15 km are corresponding to 1113 K and 720 K, respectively. In light of temperature gradient, the depth of 15 km can't reach a relatively high temperature of 1113 K. Therefore, the partial melting of gneiss is impossible to occur at the depth of 15 km with a relatively lower temperature of 720 K (Dai et al., 2014; Fuji-ta et al., 2007). Just described by Xiao et al. (2007), a large amount of granites is outcropped in the Yanshanian intrusive rocks of Dabie-Sulu ultra-high pressure metamorphic belt, and the granites are formed in the deeply upper mantle. Previously reported results from field magnetotelluric data have already disclosed that the high conductivity anomalies in the Dabie-Sulu orogen are interpreted as the cause of the partial melting and water-bearing (or saline-bearing) fluids.

**Response to the anonymous Reviewer 3#:**

After a careful reading of the revised manuscript, I found that my suggested corrections were taken into account and incorporated properly and thus, the overall picture improved significantly, both in scientific terms and in terms of the quality of the English language. I notice 2 minor changes that should be taken into consideration: 1. First, the authors should give the equation for the calculation of activation volumes (line 303).

Thanks for your valuable comments. According to your precious comments, we have already supplemented the activation energy values at the pressure range of 0.5 to 2.0 GPa in Table 3 and the calculated equation in the revised manuscript.

2. Second, in Table 3, the authors should explain the parameters, and change the symbol of the correlation coefficient to R. I suggest that the revised manuscript will make a significant contribution to the field of geophysical implications of laboratory conductivity measurements and it is suitable for publication in Solid Earth.

Thanks for your valuable comments and suggestions. The fitted equation and explanation of the parameters have been already supplemented in the revised manuscript. Meantime, the symbol of correlation coefficient of R has been corrected.

In summary, the Editor of Professor Ulrike Werban, Professor Fabrice Gaillard and two anonymous reviewers put forward many preciously constructive and enlightened comments and advisements. In the revised paper, we try my best to answer all of them and present a detailed response one by one, very carefully. Each correspondent context content, figure, table and reference has been rechecked and reedited very carefully on the base of the officially announced publication format from the journal website of Solid Earth. In here, please accept my most honest greetings and thanks for my own heart to the Editor of Professor (Professor Ulrike Werban), Professor Fabrice Gaillard and two anonymous reviewers for their hard work in completing conscientious comments. At current, we think that a thoroughly substantial and great improvements have been made for the revised manuscript, and hope it is now acceptable for publication in *Solid Earth*.

With best Regards,

Lidong Dai, PhD, Corresponding author